# The value of wake steering wind farm flow control in U.S. energy markets

Eric Simley[1], Dev Millstein[2], Seongeun Jeong[2], and Paul Fleming[1]

[1]National Wind Technology Center, National Renewable Energy Laboratory, Golden, CO 80401, USA
[2]Lawrence Berkeley National Laboratory, Berkeley, CA 94720, USA

*Correspondence to*: Eric Simley (eric.simley@nrel.gov) and Dev Millstein (dmillstein@lbl.gov)

**Abstract.** Wind farm flow control represents a category of control strategies for achieving wind plant-level objectives, such as increasing wind plant power production and/or reducing structural loads, by mitigating the impact of wake interactions between wind turbines. Wake steering is a wind farm flow control technology in which specific turbines are misaligned with the wind to deflect their wakes away from downstream turbines, thus increasing overall wind plant power production. In addition to promising results from simulation studies, wake steering has been shown to successfully increase energy production through several recent field trials. However, to better understand the benefits of wind farm flow control strategies such as wake steering, the *value* of the additional energy to the electrical grid should be evaluated—for example, by considering the price of electricity when the additional energy is produced. In this study, we investigate the potential for wake steering to increase the value of wind plant energy production by combining model predictions of power gains using the FLOw Redirection and Induction in Steady State (FLORIS) engineering wind farm flow control tool with historical electricity price data for 15 existing U.S. wind plants in four different electricity market regions. Specifically, for each wind plant, we use FLORIS to estimate power gains from wake steering for a time series of hourly wind speeds and wind directions spanning the years 2018–2020, obtained from the ERA5 reanalysis data set. The modeled power gains are then correlated with hourly electricity prices for the nearest transmission node. Through this process we find that wake steering increases annual energy production (AEP) between 0.4% and 1.7%, depending on the wind plant, with average increases in potential annual revenue (i.e., annual revenue of production (ARP)) 4% higher than the AEP gains. For most wind plants, ARP gain was found to exceed AEP gain. But the ratio between ARP gain and AEP gain is greater for wind plants in regions with high wind penetration because electricity prices tend to be relatively higher during periods with below-rated wind plant power production, when wake losses occur and wake steering is active; for wind plants in the Southwest Power Pool—the region with the highest wind penetration analyzed (31%)—the increase in ARP from wake steering is 11% higher than the AEP gain. Consequently, we expect the value of wake steering, and other types of wind farm flow control, to increase as wind penetration continues to grow.

# 1 Introduction

Wind farm flow control is a technology that coordinates the control actions of individual turbines to achieve wind plant-level
objectives, such as increasing overall energy production or reducing structural loads, by influencing aerodynamic interactions
between turbines (Boersma et al., 2017; Meyers et al., 2022). Wake steering is a type of wind farm flow control in which
upstream wind turbines are intentionally misaligned with the wind, thereby deflecting their wakes away from downstream
turbines to increase overall wind plant power production, despite the power loss incurred by the misaligned turbines (Dahlberg
and Medici, 2003; Wagenaar et al., 2012; Boersma et al., 2017; Meyers et al., 2022). A spate of recent field trials performed
on small subsets of wind turbines at commercial wind plants have confirmed the ability of wake steering to increase energy
production in realistic scenarios (Fleming et al., 2020; Doekemeijer et al., 2021; Simley et al., 2021; Howland et al., 2022).

An analysis of the potential annual energy production (AEP) increases from wake steering for existing U.S. wind plants
conducted by Bensason et al. (2021) revealed estimated AEP gains between 0.24% and 3.17% for a representative set of 60
wind plants, with an average AEP gain of 0.8%. The authors found that the potential AEP gains from wake steering are strongly
correlated with the magnitude of the wake losses suffered by a wind plant, which are determined largely by the wind plant
layout and wind direction distribution at the site. In this study we aim to augment estimates of AEP gain by considering the
*value* of the additional energy to the grid at the location and time it is added, quantified as the additional revenue generated.
For this analysis to be meaningful it is important to use realistic market prices. Pricing within the electricity system is volatile
(in most regions hourly prices typically span multiple orders of magnitudes), and pricing patterns vary year to year and region
to region. This volatility in pricing and electricity markets in general suggests that careful analysis is required to understand
the additional revenue that can be achieved with wind farm flow control, as total revenue may be sensitive to the ability to
provide energy gains during a subset of high-priced hours. An initial assessment of the impact of wind farm flow control on
the value of the electricity generated was performed by Kölle et al. (2022). The authors evaluated the energy and revenue
increases for different wind turbine and wind farm flow control strategies using two reference offshore wind plants, one year
of simulated wind speed and wind direction time series off the west coast of Denmark, and corresponding simulated hourly
electricity price time series representing market scenarios in 2020 and 2030. In general, the authors found that the revenue
increase for a particular wind plant depends strongly on the distribution of electricity prices for different wind speeds and
directions, the power increase from wind farm flow control for each wind condition, as well as the type of wind farm flow
control strategy used. Further, Kölle et al. (2022) optimized the control system of a single wind turbine considering revenue
and structural loads in the objective function, finding that revenue could be boosted without increasing structural loads by
generating more than the turbine's intended rated power when electricity prices are high and derating the turbine to reduce
damage when prices are low.

Further complicating the analysis of the additional revenue from wind farm flow control is that pricing patterns are impacted
by wind generation itself, and prices are more strongly tied to wind generation as wind power accounts for a larger portion of
total generation within a region (Seel et al., 2021; Swisher et al., 2022; Millstein et al., 2021; Prol et al., 2020; Brown and

Sullivan, 2020; Loth et al., 2022). Energy prices typically decrease in a region during hours with substantial wind generation, so energy gain during these hours would provide little value. On the other hand, wind farm flow control may be most valuable in a region during hours with relatively little wind generation, as prices may be more likely to be higher during low wind hours. In this analysis we explore the interplay between regional wind penetration and the hours in which wind farm flow control is most valuable by examining the increase in revenue from wind farm flow control in regions with low and high wind penetration. Specifically, we model wake steering wind farm flow control across 15 wind plants spanning four different electricity system regions in the United States. The four regions contain different levels of wind generation and include the two U.S. regions with the highest portion of electricity generated from wind power, as well as regions with smaller levels of wind generation. We pair this wake steering modeling with local, empirical hourly pricing patterns spanning three years, 2018 through 2020. We then explore trends across different time spans and regions to gain insight into what drives the revenue potential of wind farm flow control, and how it might change with additional wind deployment.

The rest of this article is organized as follows. In Section 2, we provide an overview of the 15 wind plants investigated. Section 3 discusses the models used in the analysis, including NREL's FLOw Redirection and Induction in Steady State (FLORIS) engineering wind farm flow control modeling tool (NREL, 2022), the process for modeling existing U.S. wind plants, and the ERA5 reanalysis wind resource data set (Hersbach et al., 2020). We describe the methods for optimizing wake steering control as well as estimating AEP and revenue gain in Section 4. In Section 5, we present the results of the study by comparing the AEP and potential revenue gains for the different wind plants and regions, assessing the dependence of the energy and revenue gains on wind speed and time of day, and examining the tendency for revenue gains to be concentrated in relatively short periods of time during the year. Section 6 concludes the article with a discussion of the results and suggestions for further research.

## 2 Overview of wind plants investigated

We selected wind plants from an area spanning much of the center of the country and connected to four separate electricity market regions (Fig. 1). Market regions are determined by the Independent System Operator (ISO) or Regional Transmission Operator (RTO), also shown in Fig. 1. For simplicity we will refer to all regions as ISO regions, though some of them are RTOs; the difference between the designations is unimportant for our purposes. We included wind plants in the Midcontinent Independent System Operator (MISO), the Southwest Power Pool (SPP), the Electric Reliability Council of Texas (ERCOT), and the PJM RTO. It is important to include plants from different regions, as each region has a somewhat unique pattern of wholesale electricity pricing. Importantly, regional pricing patterns are influenced by wind deployment levels, which could impact the value assessment of wind farm flow control. Therefore, we include wind plants in regions in which wind accounts for a substantial portion of generation, such as 31% in SPP and 23% in ERCOT, but also from regions where wind accounts for a smaller portion of total generation, such as 11% in MISO and 3% in PJM (all in the year 2020).

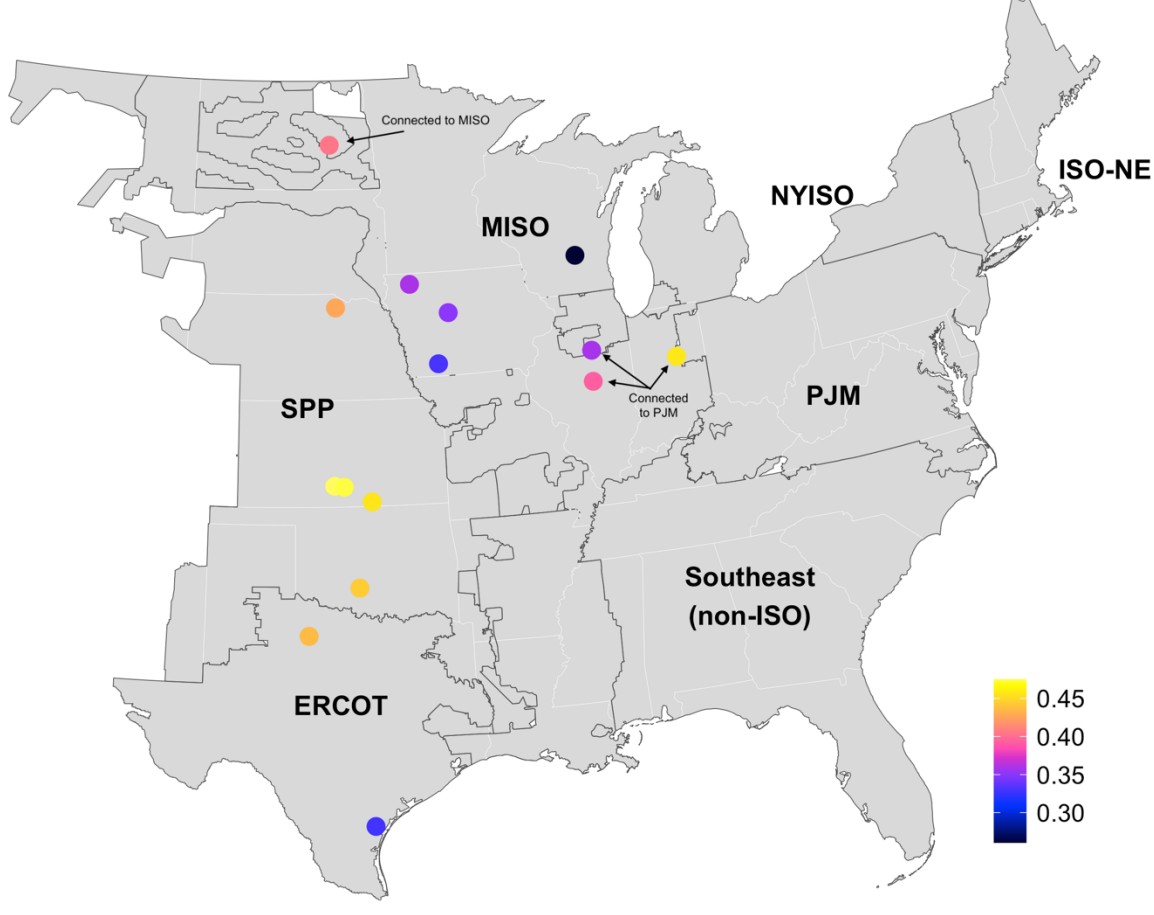

**Figure 1:** Wind plants in the study sample, which are located in the center of the country across four different ISO regions: ERCOT, SPP, MISO, and PJM. Each plant's recorded average capacity factor over 2018–2020 is shown by color. Plants connect to the ISO in which they are located unless otherwise labeled (labels are also included where plant region is ambiguous).

We selected plants with the following criteria: large plants with many turbines (minimum size was 150 MW), relatively new plants (with the oldest completed in 2009 and most completed during or after 2014), relatively isolated plants to minimize the potential for inter-plant wake effects (we do not model inter-plant wakes in this study), and plants across many different locations within each of the ISO regions. The plants we picked had recorded capacity factors that ranged from 0.26 to 0.48 averaged over the study period (Fig. 1). Details on plant configuration can be found in Table 1 and were derived from the United States Wind Turbine Database (Hoen et al., 2018); additional details were derived from the Land-Based Wind Market Report (LBWMR) (Wiser et al., 2022). Annual recorded generation was derived from the U.S. Energy Information Administration Form 923 (U.S. Energy Information Administration, 2021).

**Table 1:** Description of wind plants in the study sample (EIA: Energy Information Administration; COD: Commercial Operation Date).

| Plant Name | ISO | EIA ID | Latitude (°) | Longitude (°) | Capacity (MW) | COD Year | Average Capacity Factor | Turbine Rated Power (MW) | Turbine Rotor Diameter (m) |
|---|---|---|---|---|---|---|---|---|---|
| Papalote Creek I | ERCOT | 56983 | 27.935 | -97.463 | 180 | 2009 | 0.33 | 1.65 | 82 |
| Glacier Hills | MISO | 57199 | 43.577 | -89.112 | 162 | 2011 | 0.26 | 1.8 | 90 |
| Minonk Wind Farm | PJM | 57284 | 40.872 | -88.950 | 200 | 2012 | 0.36 | 2.0 | 90 |
| Wildcat Wind Farm I | PJM | 57862 | 40.340 | -85.864 | 200 | 2012 | 0.46 | 1.6 | 100 |
| Courtenay Wind Farm | MISO | 58658 | 47.190 | -98.604 | 200 | 2016 | 0.40 | 2.0 | 100 |
| Grande Prairie Wind Farm | SPP | 58695 | 42.611 | -98.470 | 400 | 2016 | 0.43 | 2.0 | 110 |
| Lundgren Wind Project | MISO | 58884 | 42.338 | -94.185 | 251 | 2014 | 0.35 | 2.346 | 108 |
| Radford's Run Wind Farm | PJM | 59061 | 40.000 | -89.037 | 306 | 2017 | 0.39 | 2.0 | 110 |
| Adams Wind | MISO | 59637 | 40.921 | -94.675 | 154 | 2015 | 0.33 | 2.346 2.415 | 108 |
| Slate Creek Wind Project | SPP | 59837 | 37.123 | -97.295 | 150 | 2015 | 0.46 | 2.0 | 110 |
| O'Brien Wind | MISO | 60326 | 43.199 | -95.599 | 250 | 2016 | 0.36 | 2.346 2.415 | 108 |
| Horse Creek Wind Farm | ERCOT | 60339 | 33.358 | -99.541 | 230 | 2016 | 0.44 | 2.3 | 116 |
| Rush Springs Wind | SPP | 60592 | 34.700 | -97.804 | 249 | 2016 | 0.44 | 2.075 | 116 |
| Ninnescah Wind Energy | SPP | 60620 | 37.588 | -98.610 | 208 | 2016 | 0.48 | 1.79 1.715 | 100 103 |
| Kingman Wind | SPP | 60639 | 37.556 | -98.271 | 207 | 2016 | 0.47 | 1.79 1.715 | 100 103 |

**3 Models**

In this section, we discuss the models used to estimate energy production for the wind plants investigated, with and without wake steering control, as well as the process for modeling the historical wind speed and direction time series at each wind plant.

### 3.1 FLORIS wind farm flow control engineering tool

Wind plant energy production is modeled using the open-source FLORIS engineering wake modeling software framework for the design and analysis of wind farm flow controllers (NREL, 2022). FLORIS models wind plant power production for different inflow conditions (i.e., wind speed, wind direction, turbulence intensity, air density, wind shear, and wind veer) using a simple wind turbine model consisting of the turbines' coefficients of power and thrust as a function of wind speed, hub heights (the height of the top of the tower supporting the wind turbine), and rotor diameters. Wake interactions, including wake

redirection resulting from yaw misalignment, are computed using computationally efficient engineering wake models.

    In this study, we use the default Gauss-curl hybrid (GCH) wake model in FLORIS (King et al., 2021). The GCH model is based on the analytical self-similar Gaussian wake velocity deficit model developed by Bastankhah and Porté-Agel (2014) and Niayifar and Porté-Agel (2016), and the model of wake deflection from yaw misalignment—derived using budget analysis of the Reynolds-averaged Navier-Stokes equations—proposed by Bastankhah and Porté-Agel (2016). The rate of wake

recovery/wake expansion in the Gaussian wake model is governed by the total turbulence intensity resulting from the combination of the ambient turbulence intensity and wake-added turbulence. The GCH model augments this standard Gaussian wake model by incorporating elements of the curl wake model described by Martínez-Tossas et al. (2019), which captures the effects of trailing large-scale counter-rotating vortices in the flow caused by yaw misalignment in a computationally efficient manner. Specifically, GCH models 1) yaw-added recovery, in which the trailing vortices from yaw misalignment increase

wake recovery by enhancing wake mixing with the ambient flow, and 2) secondary steering, wherein the vortices continue to deflect the wakes of downstream turbines operating in the wake of a yawed turbine. Note that the secondary steering effect could also be partially explained by changes in the effective wind direction at waked wind turbines because of non-uniform inflow (Schepers et al., 2012). To model the superposition of multiple wakes at a given wind turbine location, we use the sum-of-squares of the velocity deficits from multiple upstream turbines relative to the freestream velocity. Note that in FLORIS

V2.5 (NREL, 2022), used in this study, we implement the above-mentioned models using the "gauss_legacy" velocity deficit model, "gauss" deflection model, "crespo_hernandez" wake-added turbulence model, and "sosfs" wake combination model, with the default parameters. Lastly, the power loss from yaw misalignment is modeled using the approach suggested by Bossanyi (2019) by scaling the rotor effective wind speed by the factor $(\cos \gamma)^{p_p/3}$, where $\gamma$ is the yaw misalignment and $p_p$ is a tunable cosine exponent. For this study, we chose the commonly used value of $p_p = 2$ (Medici, 2005; Howland et al.,

2020).

    Wake losses as well as the energy gain possible with wake steering depend strongly on turbulence intensity (Bensason et al., 2021; Simley et al., 2022); both wake losses and the potential increases in wind plant power production from wake steering are higher when turbulence is low. However, the ERA5 wind resource data set used to determine time series of hourly wind speeds and directions for each wind plant in this study (which will be discussed in Section 3.3) does not contain a measure of

turbulence. To address this limitation while still capturing the impact of time-varying turbulence intensities in FLORIS, we model turbulence intensity (TI) as a function of wind speed following the normal turbulence model definition in the IEC 61400-

1 wind turbine design standard, in which turbulence decreases as wind speed increases (International Electrotechnical Commission, 2005):

$$\text{TI}(\bar{u}) = I_{\text{Ref}}\left(0.75 + \frac{5.6}{\bar{u}}\right), \tag{1}$$

where $\bar{u}$ is the mean wind speed—given by the hourly wind speed from the ERA5 data set in this work—and $I_{\text{Ref}}$ is intended
to be the expected value of the turbulence intensity at 15 m/s. But in this study, we treat $I_{\text{Ref}}$ as a tuning parameter to achieve desired average annual wake losses at each wind plant. Specifically, Fleming et al. (2020) and Fleming et al. (2021) found that when using a turbulence intensity of 8%–10%, FLORIS predictions closely matched the average wake losses experienced by a pair of wind turbines at commercial wind plants; therefore, we tune $I_{\text{Ref}}$ for each wind plant so that the average annual wake losses predicted using FLORIS match those based on a contact turbulence intensity of 9%.

An example of a flow field modeled using FLORIS for the Horse Creek wind plant investigated in this study is provided in Fig. 2 for a mean wind speed of 8 m/s and turbulence intensity of 8.75% (determined using Eq. (1)). The flow around a subset of wind turbines in the wind plant is compared using baseline control with the turbines oriented into the wind (Fig. 2b) and using wake steering control (Fig. 2c), with optimal yaw offsets determined by FLORIS (yaw offset optimization will be discussed further in Section 4.1).

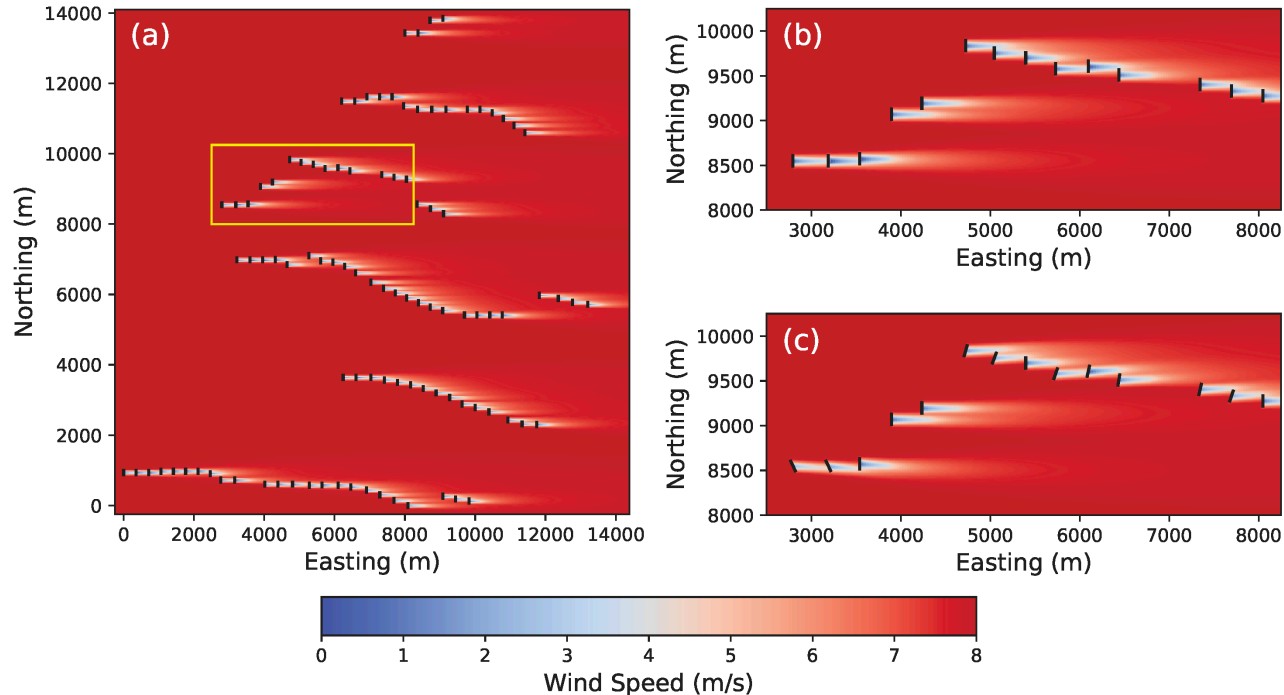

**Figure 2:** Flow fields computed using FLORIS for the Horse Creek Wind Farm wind plant for a wind direction of 270° (i.e., westerly flow), wind speed of 8 m/s, and turbulence intensity of 8.75% for (a) the entire wind plant with baseline control, and the region encompassed by the yellow rectangle in Fig. 2a with (b) baseline control and (c) wake steering control using optimal yaw offsets determined by FLORIS.

## 3.2 Wind plant and wind turbine modeling

For each wind plant analyzed, a FLORIS model is created using information published in the United States Wind Turbine Database (Hoen et al., 2022). Specifically, the latitude and longitude for each turbine are used to define the wind plant layout (e.g., see Fig. 2), and the hub heights, rotor diameters, and rated power values listed for each turbine are used to specify the turbine properties. FLORIS requires the hub height, rotor diameter, and the coefficient of power ($C_P$) and coefficient of thrust ($C_T$) curves as a function of wind speed for each turbine. Whereas the hub height and rotor diameter parameters are directly used in the FLORIS model, we estimate the $C_P$ and $C_T$ curves based on the corresponding curves for the IEA 3.4 MW reference wind turbine (RWT) model (Bortolotti et al., 2019; IEA Wind Task 37, 2021), with a rotor diameter of $D = 130$ m, using the approach developed by Bensason et al. (2021).

To estimate the $C_P$ and $C_T$ curves for an arbitrary wind turbine, we assume that the curves are the same as those published for the IEA 3.4 MW RWT, except that they are scaled as a function of wind speed, so that rated power is reached at the estimated rated wind speed of the turbine of interest rather than the rated wind speed of 9.8 m/s for the IEA 3.4 MW RWT (Bortolotti et al., 2019). Following a similar procedure as Bensason et al. (2021), rated wind speed is estimated for the turbine of interest using the standard power equation

$$P = \frac{1}{2}\rho A C_P u^3, \tag{2}$$

where $P$ represents power, $\rho$ is the standard air density of 1.225 kg/m$^3$, $A$ indicates the rotor area, and $u$ is the rotor effective wind speed. Assuming the value of $C_P$ at the rated wind speed is equal to the coefficient of power of the IEA 3.4 MW RWT at its rated wind speed, $C_{P,Rated,Ref} = 0.439$, the rated wind speed is estimated by solving Eq. (2) for the wind speed at which the turbine produces rated power:

$$\hat{u}_{Rated} = \left(\frac{2P_{Rated}}{\rho A C_{P,Rated,Ref}}\right)^{\frac{1}{3}}. \tag{3}$$

Next, wind speeds $u$ for the turbine of interest that are greater than the cut-in wind speed—which is assumed to be equal to the cut-in wind speed for the IEA 3.4 MW RWT of $u_{Cut-in,Ref} = 3$ m/s—are mapped to equivalent wind speeds for the IEA 3.4 MW RWT, $u_{Ref}$, which correspond to the same fraction of the wind speed range between $u_{Cut-in,Ref}$ and rated wind speed (wind speeds below $u_{Cut-in,Ref}$, for which the turbine is not producing power, are simply mapped to the same wind speed):

$$u_{Ref} = \begin{cases} \frac{\left(u - u_{Cut-in,Ref}\right)\left(u_{Rated,Ref} - u_{Cut-in,Ref}\right)}{\hat{u}_{Rated} - u_{Cut-in,Ref}} + u_{Cut-in,Ref}, & u \geq u_{Cut-in,Ref} \\ u, & u < u_{Cut-in,Ref} \end{cases} \tag{4}$$

Lastly, the values of $C_P$ and $C_T$ for the turbine of interest at wind speed $u$ are treated as the $C_P$ and $C_T$ values for the IEA 3.4 MW RWT corresponding to wind speed $u_{Ref}$.

To illustrate the method for estimating $C_P$ and $C_T$ curves described in this section, the resulting power and $C_T$ curves for the wind turbines at the Rush Springs Wind plant investigated in this study, with a rated power of 2.075 MW and rotor diameter of $D = 116$ m, are compared to the reference curves for the IEA 3.4 MW RWT in Fig. 3. Note that for the wind turbines at Rush Springs, Eq. (3) yields an estimated rated wind speed of 9 m/s, which is less than the rated wind speed of 9.8 m/s for the IEA 3.4 MW RWT, effectively shrinking the wind speed dependence of the estimated $C_P$ and $C_T$ curves compared to the

reference curves. Across all 15 wind plants, the estimated mean wind speeds of the FLORIS wind turbine models using Eq. (3) range from 9 m/s to 10.5 m/s, with a mean value 9.6 m/s.

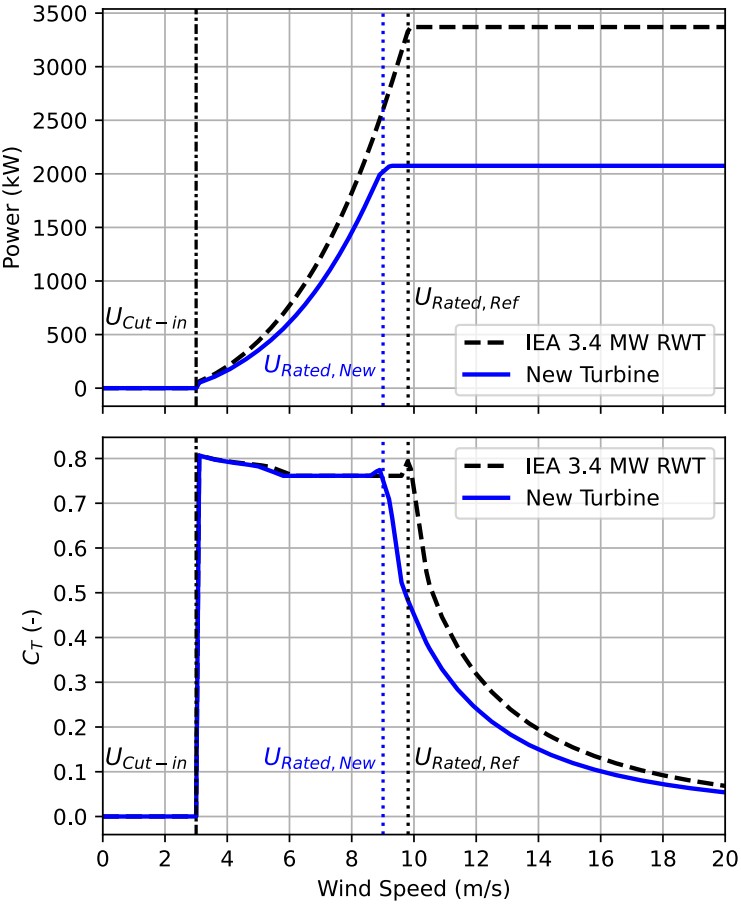

**Figure 3:** (a) Power curves and (b) coefficient of thrust ($C_t$) curves for the IEA 3.4 MW reference wind turbine (RWT) model (Bortolotti et al., 2019) and a new wind turbine model representing the 2.075 MW wind turbines at the Rush Springs Wind plant with rotor diameter $D = $ 116 m.

### 3.3 Wind resource modeling

Wind speed and direction observational data measured at hub height (typically 80–100 m above ground) are not publicly available (Kusiak, 2016; Archer et al., 2014). Therefore, we rely on modeled wind speeds. The purpose here is to develop an hourly wind speed and direction time series at each wind plant with which to input into the detailed wind plant models. Though we cannot perfectly characterize wind speeds, our goal is to at least roughly characterize variation in hourly wind speed and direction, as the wake modeling is most sensitive to those two inputs.

Hourly wind speed and direction are based on the reanalysis model ERA5 (Hersbach et al., 2020). When comparing across two publicly available reanalysis data sets commonly used in wind power applications, ERA5 has been found to outperform MERRA2 (Gelaro et al., 2017). For example, Olauson (2018) found that across Sweden, ERA5 had better performance than MERRA2 for both mean error and correlation metrics. Davidson and Millstein (2022) found similar results in Texas, and also showed that Pearson correlation coefficients between modeled and reported hourly generation were strongest during daytime and during winter months (>0.85) and weakest during the summer nighttime (~0.72). Though validation across additional regions would be useful, it is not currently available, so we proceeded with using ERA5 wind data as a basis for our modeling.

In the case of wind speed, we debiased the model outputs using reported hourly wind generation data. The challenge is that in most regions, hourly generation records are only available after aggregation across the region (reported by each ISO/RTO), while only monthly generation records are available at each plant (reported by the U.S. Energy Information Administration Form 923 (U.S. Energy Information Administration, 2021)). An exception is that in ERCOT, plant-level hourly generation data are available. To debias wind speeds we followed the approach applied in Wiser et al. (2021) but extended the approach to cover years 2018–2020. Variations of this approach have been applied in Millstein et al. (2021) and Wiser et al. (2022).

The details of the debiasing process can be found in the aforementioned citation; however, the approach is summarized here (note that there is no debiasing process for wind direction; wind direction is taken based solely on the raw ERA5 hub height wind products). Broadly, we use an iterative process to scale raw modeled wind speeds so that the derived generation estimates match recorded generation at both the hourly regional level and monthly plant level. Specifically, we first estimate plant-level hourly generation using a wind-plant-appropriate power curve and ERA5 wind speed found at the turbines' hub height. Generation is combined across the region and compared to the regional hourly total. Generation across all wind plants is scaled in each hour so that estimated generation matches reported regional generation. Wind plant capacity limitations are maintained (i.e., no plant is allowed to output more than its nameplate capacity in any given hour). An additional scaling is then applied to each wind plant separately so that modeled plant generation matches reported plant generation on the monthly scale. These scaling steps are repeated in an iterative process until convergence is found—that is, modeled generation matches both regionwide hourly generation records and plant-specific monthly generation records. Finally, wind speed is backed out of the debiased generation estimates using the plant-specific power curves. The result is a set of hourly wind speeds that after the application of simple power curve modeling would roughly reproduce the available generation records. Note that prior to beginning this iterative process we adjust generation records for curtailment (i.e., we compare to recorded generation prior to

215 reductions due to curtailment). Finally, we can use a simpler process for plants in ERCOT because plant-level hourly
generation records are available, and we can back out wind speed directly from these plant-level records.

## 4 Methods

The methods used to determine the potential increases in energy and revenue with wake steering control are briefly described
in this section.

### 4.1 Wake steering optimization using FLORIS

To determine energy and revenue gains from wake steering for each wind plant, we find the optimal yaw offsets for each wind
turbine that maximize wind plant power production as a function of wind direction, in 1° steps, and wind speed, in 0.5-m/s
steps from 3 to 25 m/s. Note that in this study, the optimal yaw offsets for power maximization and revenue maximization are
equivalent because we do not consider the potential impacts of wake steering on expenditures such as operations and
225 maintenance costs during the lifetime of the wind plants. To avoid extreme yaw misalignments, we constrain the lower and
upper yaw offset bounds to −25° and +25°, respectively. Note that in practice, further reducing the magnitude of the yaw offset
bounds as wind speed increases may be necessary to mitigate higher structural loads caused by yaw misalignment (Damiani
et al., 2018; Shaler et al., 2022). Additionally, different bounds on the magnitude of the negative and positive yaw offset
bounds may be appropriate to address the asymmetry in structural loading from yaw misalignment. However, in this study, we
simply limit the yaw offset magnitude to 25° for all wind conditions to estimate *potential* power increases from wake steering.

For each wind direction and wind speed bin, yaw offsets are optimized using the Serial-Refine (SR) optimization method in
FLORIS (Fleming et al., 2022). As discussed by Fleming et al. (2022), compared to the gradient-based sequential least-squares
programming (SLSQP) optimization method implemented in FLORIS using the SciPy Python package (Virtanen et al.,
2020)—which was used to estimate the AEP gain from wake steering for the representative set of 60 U.S. wind plants by
235 Bensason et al. (2021)—the SR method tends to find yaw offsets that yield slightly higher power production while requiring
significantly less computation time. The SR method begins by stepping serially through each wind turbine in a wind plant,
from the most upstream to the farthest downstream turbine, and evaluating the power produced by the wind plant for a discrete
set of $N_{Yaw}$ yaw offsets evenly spaced between the lower and upper offset bounds (we used $N_{Yaw} = 5$ in this study, resulting in
the set: {0°, ±12.5°, and ±25°}). For each turbine, the yaw offset that maximizes wind plant power production is identified and
240 assigned to the turbine while yaw offsets for the remaining downstream wind turbines are evaluated. After the optimal yaw
offsets are identified from this first coarse search, the process is repeated using a refined set of $N_{Yaw}$ yaw offset candidates
centered on the coarse optimal offset. For example, if an optimal yaw offset of 12.5° is found during the coarse search for a
particular turbine, offsets of 6.25°, 9.375°, 12.5°, 15.625°, and 18.75° will be evaluated during the refined search. Finally, the
SLSQP optimization method, limited to only 10 iterations to manage computation time, is used to further refine the optimal

offsets, treating the optimal offsets from the previous step as the initial conditions. Examples of the optimal yaw offsets found using the SR method described here are shown in Fig. 2c for a subset of turbines in the Horse Creek Wind Farm wind plant.

## 4.2 Energy and revenue gain estimation

We estimate the increase in AEP that can be achieved through wake steering by comparing the modeled annual energy produced with wake steering optimization to the annual energy produced with no wake steering adjustments, where the annual energy is computed as the sum of the modeled energy production over all 1-hour periods in a specific year. We determine the energy produced by a wind plant, with and without wake steering, for the specific wind direction and wind speed corresponding to a given hour, by linearly interpolating the precomputed tables of FLORIS results with 1° wind direction resolution and 0.5 m/s wind speed resolution discussed in Section 4.1. A parallel process is followed to determine the gain in annual revenue of production (ARP), which is calculated by summing annual time series of hourly energy production multiplied by the hourly electricity prices at electricity nodes near each wind plant (Hitachi, 2022) for baseline and wake steering control. One difference, however, is that all negative prices are set to zero. Since ARP is calculated as the sum of hourly prices multiplied by hourly energy output, setting the price to zero for negatively priced hours simulates the curtailment of production during those hours. Thus, any possible wake steering controls during those hours have no impact on our ARP gain calculation. In practice, there are often contractual reasons for a plant to avoid curtailing its output during negatively priced hours, but the contractual considerations are beyond the scope of this analysis. Note that ARP is similar to the "income gain" reported by Kölle et al. (2022) and the "energy value" (EV) metric presented by Millstein et al. (2021), except ARP is expressed in units of $/year, whereas EV represents the revenue per unit of energy produced ($/MWh).

By using hourly wind data from ERA5 to assess wake steering in this study, we are missing information about higher frequency variations in wind speed and direction, which could impact the potential increase in energy from wake steering for any particular hour. For example, if the distribution of wind directions within a specific hour leads to fewer wake interactions than the single hourly wind direction, the energy gain could be lower for that hour. But if the actual distribution of wind directions causes greater wake losses than the single hourly wind direction, the energy gain from wake steering could be enhanced. Therefore, we expect that the impacts of sub-hourly variations in wind conditions will tend to cancel out over the annual periods investigated without significantly affecting the results.

Note that, in practice, the energy and revenue gains from wake steering are expected to be lower than those predicted by FLORIS because of the inability of wind turbines to perfectly adapt their yaw offsets to the optimal values in realistic time-varying wind conditions (see Simley et al. (2020) and Fleming et al. (2020) for a deeper discussion of this limitation). However, in the present study, our goal is to assess the potential impact of wake steering on AEP and ARP gain, rather than model specific controller limitations, which we expect to be turbine-specific and likely to improve as wake steering technology matures.

## 5 Results

In this section, we begin by comparing the increase in AEP and ARP from wake steering control for the 15 wind plants and four electricity market regions in Section 5.1. The dependence of the increase in AEP and ARP on wind speed as well as time of day are compared in Sections 5.2 and 5.3, respectively. Lastly, Section 5.4 highlights the concentration of the overall energy and revenue gains in relatively short periods of time.

### 5.1 Annual energy and revenue gains from wake steering

Wake steering can help reduce wake losses, and this is typically assessed as a gain in annual energy production (AEP). The magnitude of wake losses at a wind plant depends on a number of factors including plant layout, distribution of wind directions and speeds, and turbine design. Due to this complexity, wake losses vary substantially by wind plant. For example, Clifton et al. (2016) describe and quantify multiple types of typical wind energy losses, finding that wake losses range from negligible to ~10% (with higher values observed for offshore wind plants (Pryor, 2021)), depending on the plant. At 10%, wake losses might be the single largest source of energy loss at a wind plant. Of course, the potential for AEP gain from wake steering is closely correlated with total wake losses. The correlation between AEP gain and total wind plant wake losses is shown in Fig. 4, which shows plant-by-plant results for each year from 2018 to 2020 and for the 15 wind plants investigated (45 separate points). Across these plants we estimate that uncontrolled wake losses range from approximately 4% to 20%, and the potential AEP gain due to wake steering ranges from 0.4% to 1.7%. The wind plants in our sample from the PJM and MISO regions tend to have higher wake losses than the sample of plants in ERCOT and to a lesser extent SPP. Wind direction in ERCOT and southern SPP tends to vary less than wind direction in other locations, which makes it easier to design plants with low wake losses.

To move from AEP gain to ARP gain we examine hourly price records at electricity nodes near each wind plant (Hitachi, 2022). We expect that the ARP of wind generation from a plant would increase by more than the AEP gain. The logic behind this expectation is as follows: 1) wake steering can increase energy generation at low and medium wind speeds, but at above-rated wind speeds, when wake interactions are negligible because of reduced rotor thrust and wind plants are already operating at full rated output, wind plants cannot increase their output with wake steering; 2) because wind speeds are regionally correlated, hours with high wind speeds will have lower electricity prices than hours with medium or low wind speeds, as high wind generation during high wind hours will reduce electricity prices with the increased supply. Therefore, we hypothesize that energy prices will tend to be relatively high during hours in which wake steering is increasing energy production, leading to ARP gains larger than AEP gains.

However, this logic is complicated by the fact that there are many factors that control energy prices and that hourly prices vary immensely over the course of the year. For example, annual average prices range between \$10/MWh and \$70/MWh depending on the region and year, but hourly prices, at an individual location, can span 3–4 orders of magnitude over the course of a single year (ranging from \$0/MWh to \$10,000/MWh). Thus, given the complexity of factors that influence

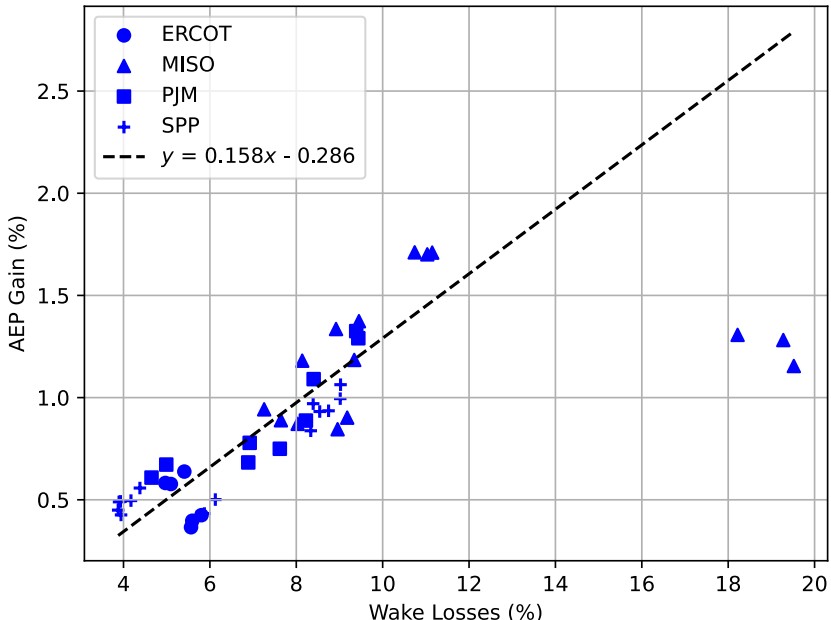

**Figure 4:** AEP gain is plotted against annual wake losses at each wind plant. Three annual values are plotted for each plant (years 2018–2020), which can often be seen grouped together, indicating variation across wind plants is much larger than variation across years.

electricity prices, and the large variation in hourly prices, it is important to actually match hourly wake steering energy gains to local prices to determine the impact on total revenue.

By matching hourly prices to hourly wake steering energy gains, we see that ARP gain is larger than AEP gain for most regions and years in our sample (see Fig. 5). Over all plants and years, average ARP gain is 4% greater than average AEP gain. ARP gain is 11% and 9% larger than AEP gain for wind plants in SPP and ERCOT, respectively, regions in which wind generation accounts for a large portion of total energy generation (see Table 2). ARP gain is only 1% larger than AEP gain in MISO and almost equivalent to AEP gain in PJM; these regions had wind penetration levels of only 11% and 3%, respectively,

in 2020.

     To reveal the impact that wind speed-dependent turbulence intensity values in FLORIS have on the estimated increases in revenue from wake steering, regional averages of AEP gain and ARP gain over all years investigated are provided in Table A1 in Appendix A for a constant turbulence intensity value of 9% in FLORIS. The AEP gains from wake steering are similar when using variable or constant turbulence intensities, but the ARP gains are significantly higher when assuming a constant

turbulence intensity. For a constant turbulence intensity of 9%, the average ARP gain over all wind plants and years is 10% greater than the average AEP gain, and the ARP gain is 21% larger than the AEP gain for SPP. As will be explained in Section 5.2, the lower ARP gains that result from more realistic wind speed-dependent turbulence intensities in FLORIS can be explained by a weaker correlation between electricity prices and the relative energy gains from wake steering, compared to the scenario with a constant turbulence intensity.

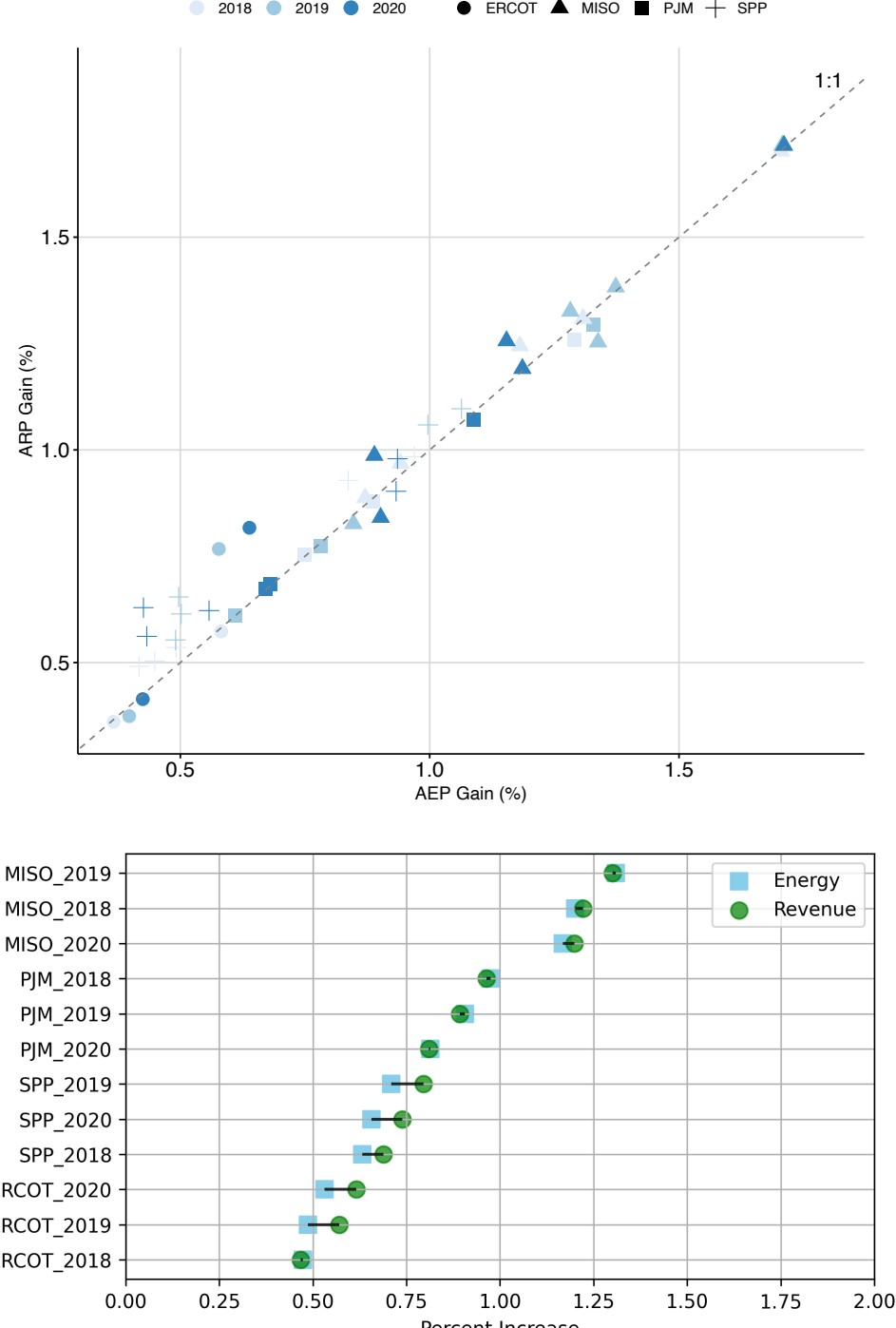

**Figure 5:** Top panel: ARP (revenue) gain plotted as a function of AEP (energy) gain; each point represents a single year for a wind plant. Bottom panel: The additional gain to ARP (revenue) above AEP (energy) gain shown as an annual regional average.

**Table 2:** Regional averages from 2018 to 2020 of AEP (energy) gain and ARP (revenue) gain, and wind penetration in 2020.

| Electricity Market Region | Energy (AEP) Gain | Revenue (ARP) Gain | ARP Gain / AEP Gain | 2020 Annual Wind Penetration |
|---|---|---|---|---|
| SPP | 0.67% | 0.74% | 1.11 | 31% |
| ERCOT | 0.50% | 0.54% | 1.09 | 23% |
| MISO | 1.23% | 1.24% | 1.01 | 11% |
| PJM | 0.89% | 0.90% | 1.00 | 3% |

## 5.2 Wind speed dependence of energy and revenue gains

One pattern that is similar across all regions is that revenue gain from wake steering control is more concentrated in low wind speed hours than is energy gain. We can see this clearly demonstrated in Fig. 6, which shows the fraction of AEP or ARP gain as a function of wind speed in each region. Focusing first on energy gain, we see the expected pattern that wake steering leads to increased energy gain with higher wind speed until wind speed moves past the turbines' rated wind speed, at which point additional controls cannot provide increased output, as the turbines are already operating at full capacity. Revenue follows this pattern to a certain extent, but the shape is shifted left—revenue gain at low wind speeds makes up a larger portion of the total revenue gain than energy gain at low wind speeds makes up of total energy gain. This observation matches the hypothesis that controls to increase power production at low wind speeds are more valuable (on a per-megawatt-hour basis) than controls at higher wind speeds (because low wind speeds are correlated with higher energy prices and high wind speeds are correlated with low energy prices). We can see in Fig. 6 that the shift of revenue towards lower wind speeds is larger in SPP, ERCOT, and MISO compared to PJM, suggesting that this effect is correlated with total wind penetration in a region. In other words, prices in regions with high wind penetration are more sensitive to wind output, so prices tend to be higher during periods of low wind output, leading controls during those periods to be more valuable. A second point here is that curtailment is also less likely to occur during periods of lower wind output, which is an additional benefit of power gains that occur during lower wind speeds.

Another trend that helps explain why the increase in ARP from wake steering tends to be higher than the AEP gain is that the *relative* increase in energy from wake steering is generally greater at lower wind speeds where electricity prices are higher. As shown in Fig. 7, which compares the relative increase in energy from wake steering (expressed as a percentage increase) to the normalized electricity prices as a function of wind speed for each region, the relative energy gains tend to be correlated with electricity prices. Namely, the relative energy gains are largest at wind speeds below ~10 m/s, when prices are generally high. Similarly, at higher wind speeds where the relative energy gains from wake steering are low (with energy gains approaching zero above the wind turbines' rated wind speeds as wake losses become negligible), regional electricity prices tend to be low as well. Note that the very large relative energy gains from wake steering at wind speeds below 5 m/s in Fig. 7 are partially caused by wake steering increasing the wind inflow at downstream turbines above the turbines' cut-in wind speeds,

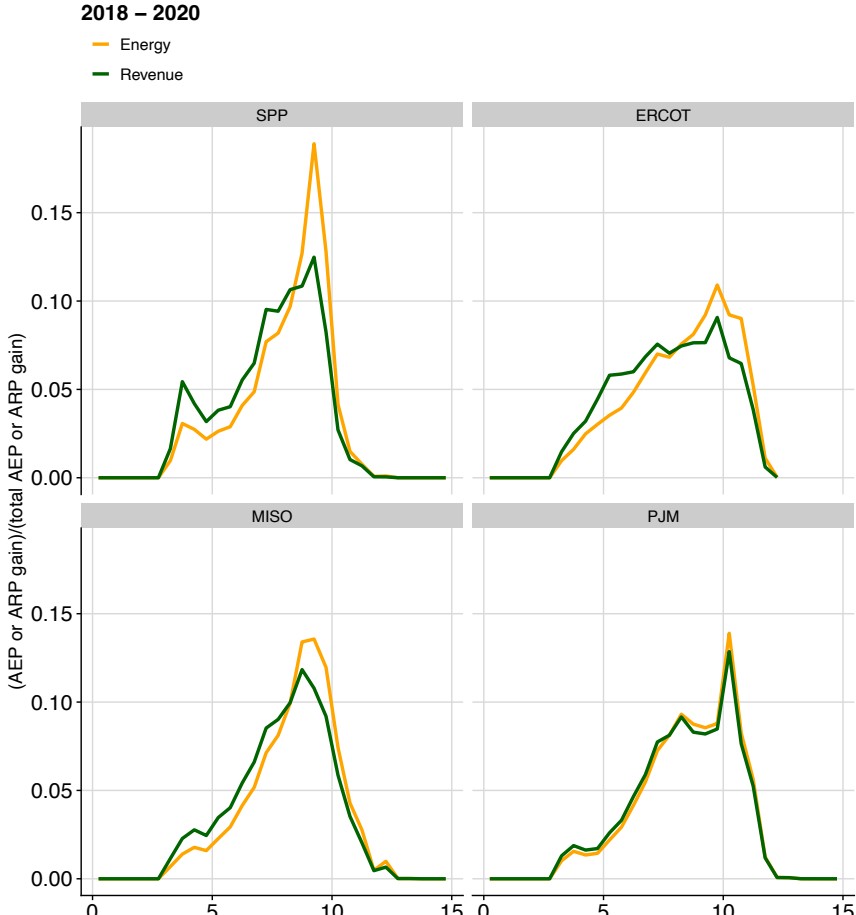

**Figure 6:** Increases in energy and revenue from wake steering control as a function of wind speed, normalized by the total AEP or ARP gain, for each region, averaged across the sample wind plants in each region and the years 2018–2020.

thus enabling them to generate *some* power instead of remaining shutdown; however, the resulting relative energy increases at these wind speeds do not translate to increases in absolute energy that are as significant, as can be seen in Fig. 6. Figure 7 also clearly shows the pattern observed in Fig. 6 whereby prices are more sensitive to wind output in regions with higher wind penetration, with the strongest correlation between prices and wind speeds occurring in SPP (the region with the highest wind penetration) and the weakest correlation appearing in PJM (where wind power makes up the lowest percentage of overall generation).

The impact of lower turbulence intensity values as wind speed increases (given by Eq. (1)) on the relative energy gain can be clearly seen in Fig. 7 for MISO and PJM for wind speeds between 5 m/s and 10 m/s. In general, during below-rated wind plant operation, as turbulence intensity decreases, wake losses increase, and wake steering becomes more effective at increasing wind plant power production. On the other hand, as shown in Fig. A1 in Appendix A, when using a constant

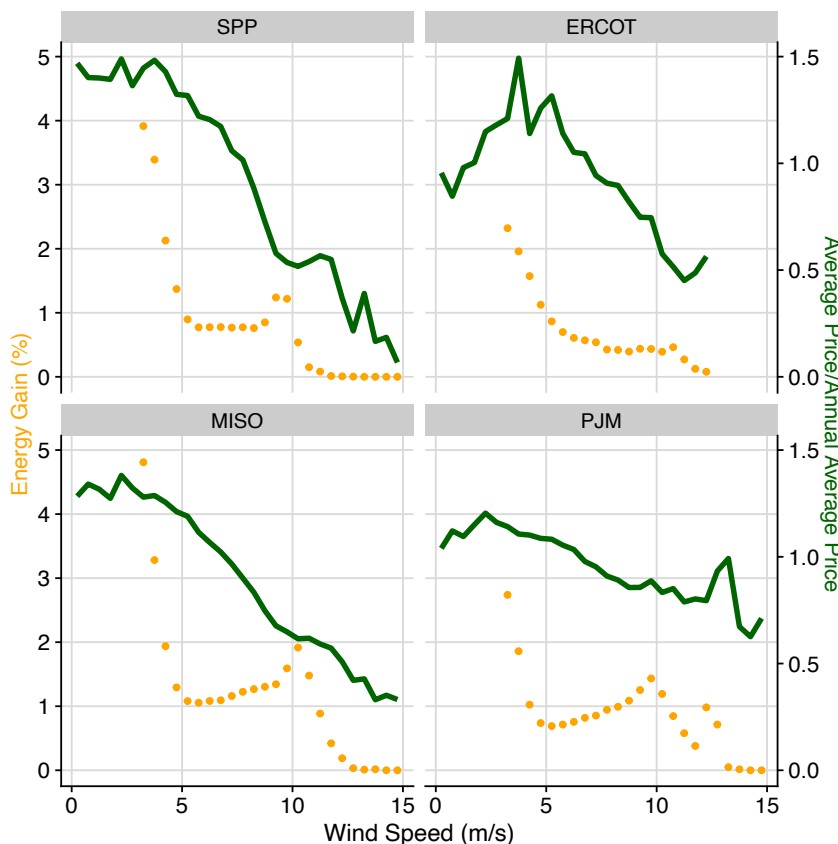

**Figure 7:** Relative energy gains and normalized electricity prices as a function of wind speed for each region. Relative energy gains are calculated independently for each 0.5 m/s wind speed bin. Average prices are also calculated independently for each wind speed bin, but then normalized by the respective annual average price across all wind speeds. Values are averaged across all wind plants in each region and the years 2018–2020. Note that despite large relative energy gains at wind speeds below 5 m/s, total energy gains at these low wind speeds are small, as shown in Fig. 6.

turbulence intensity value of 9% in FLORIS, the relative energy gains from wake steering exhibit a more consistent reduction as wind speed increases for all regions. Consequently, the correlation between relative energy gains and electricity prices is much stronger, resulting in the larger ARP gains compared to AEP gains listed in Table A1 compared to the results with variable turbulence intensity.

In summary, in addition to observing a shift in the overall revenue increases from wake steering toward lower wind speeds compared to the energy gains, we find that the lower wind speeds where electricity prices are higher tend to correspond to wind speeds where wake steering produces the greatest relative increases in energy, thus contributing to larger overall ARP gains than AEP gains.

### 5.3 Diurnal energy and revenue gain trends

There are different diurnal cycles between revenue gain and energy gain (see Fig. 8). The portion of energy gain found during nighttime is larger than the portion of energy gain during daytime. The opposite is true of revenue, which is more dependent on daytime hours than nighttime hours. These differences occur in all regions studied. Absolute energy gain peaks at night because in most locations wind generation tends to be more concentrated in nighttime, when wind speeds are typically higher, than daytime. Additionally, because wind speeds are generally higher at night, the modeled turbulence intensities using Eq.

(1) tend to be lower, which yields larger increases in power production from wake steering. In contrast, energy prices tend to be higher during the daytime in these regions. Electricity prices are a function of many factors, but most relevant here is that demand for electricity tends to be larger during the daytime (pushing daytime prices up), and wind supply tends to be higher at night (pushing nighttime prices down). The combination of these factors leads to the different diurnal patterns in energy gain and revenue gain.

### 5.4 Temporal concentration of energy and revenue gains

Both energy and revenue gains from wake steering control were found to be highly concentrated in time. Across all regions, the highest ranked individual hours of revenue gain contribute a greater portion of total ARP gain than the highest ranked hours of energy gain contributed to AEP gain. Of particular note is that the top 100 hours of revenue gain in ERCOT and SPP account for 25% and 14%, respectively, of the average annual totals (see Fig. 9). The concentration of revenue gain in a small

number of hours makes intuitive sense because electricity prices can spike by orders of magnitude above average, but there are physical limits to how much energy can be gained in any one hour.

On the other end of the spectrum, there are some hours with energy gain that contribute zero additional revenue. These hours have prices at or below zero, so the possibility of enhanced energy gain does not provide any value during these particular hours. Negative prices were particularly prevalent in SPP during 2020. More generally, hours with below average (but still

positive) prices tend to provide minimal revenue gain even with substantial energy gain.

### 6 Discussion and conclusions

Wake steering control was found to increase ARP by 0.5% to 1.2% depending on the region. These revenue gains were, on average, 4% greater than the AEP gains from wake steering, but revenue gain was discovered to be much larger than energy gain in high-wind regions such as SPP and ERCOT (where revenue gain was 11% and 9% greater than energy gain,

respectively). This conclusion helps to dispel the occasional concern that the increase in revenue from wind farm flow control will be limited because the energy gains are concentrated during nighttime hours with low prices.

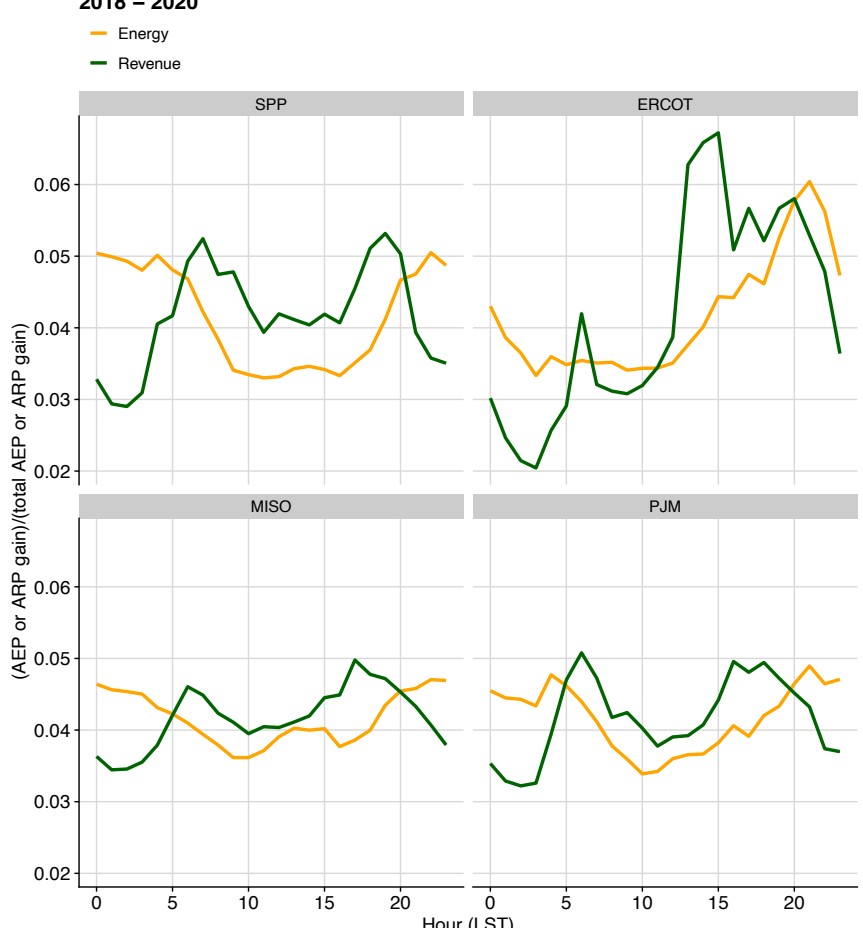

**Figure 8:** Increases in energy and revenue from wake steering control as a function of hour of the day, normalized by the total AEP or ARP gain, for each region, averaged across the sample wind plants in each region and the years 2018–2020.

Although we found that the energy gain from wake steering was concentrated during nighttime hours, the energy gain during daytime combined with relatively high daytime prices more than makes up for the lower overall energy gains during daytime hours and was sufficient to produce a majority of the ARP gain. In general, revenue gains are concentrated at lower wind speeds than energy gains, especially in high-penetration wind regions. This is likely due to regionally correlated wind patterns—that is, higher wind speed hours tend to have relatively high wind output across a region, which suppresses prices, whereas prices tend to be higher during hours of low to medium wind speed when wind plants provide less power to the grid. Additionally, the larger increases in ARP compared to AEP that were observed can be partially explained by the greater relative energy gains from wake steering at low and medium wind speeds, when electricity prices are higher, compared to higher wind speeds, where wake steering provides little or no benefit but electricity prices tend to be lower as well.

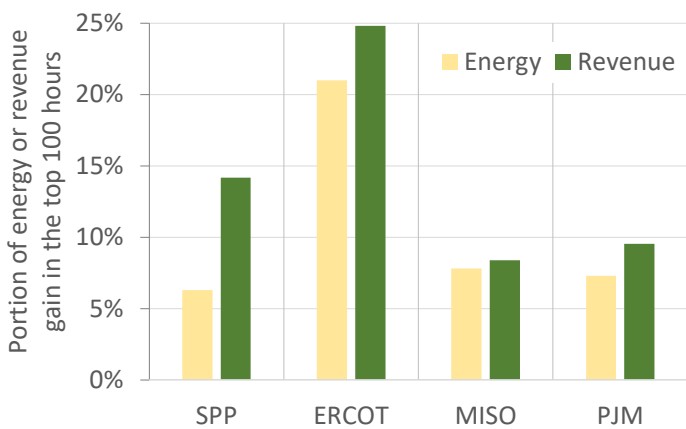

**Figure 9:** The portion of energy or revenue gain from wake steering control that can be attributed to the top 100 hours of AEP or ARP gain for each region averaged over all wind plants in the region and years 2018–2020.

Revenue gain is also driven by general volatility in the energy markets. In each region, the top 100 hours of revenue gain accounted for an outsized portion of ARP gain, ranging from 7% to 25% across the regions. These top 100 hours represent hours in which particularly high prices coincided with wake steering energy gains. Thus, we have identified two mechanisms that help drive revenue gain higher than energy gain: the first being the relatively high prices that occur during hours with low to medium wind speeds, and the second being that volatility in electricity market pricing ensures some hours produce outsized revenue gains compared to energy gains. This volatility does not have a symmetric impact on revenue—negative price spikes, which can and do occur, can be mitigated through curtailment, allowing wake steering control to capitalize on positive price spikes and simply avoid negative price spikes.

It is also important to note the limitations of this study. The key limitation is that wake steering energy gain was assessed using model processes. Imperfections in the FLORIS wake models as well as uncertainty in the estimated wind turbine coefficients of power and thrust as a function of wind speed (see Section 3.2) create uncertainty in the predicted AEP and ARP gains. Wind speeds and directions were based on ERA5 reanalysis model outputs and were input into a wake model to assess the potential for control-based energy output improvements. Though a debiasing process was applied for wind speeds, hourly wind direction was based on raw ERA5 outputs. Further, wake losses as well as the energy gains possible with wake steering strongly depend on atmospheric conditions such as stability and turbulence intensity. As discussed in Section 3.1, we attempted to capture much of the impact of time-varying turbulence intensity on AEP and ARP gains by assigning turbulence intensity in FLORIS as a function of wind speed, based on the IEC normal turbulence model definition (in which turbulence decreases as wind speed increases); however, while this models the expected average relationship between wind speed and turbulence intensity, turbulence can vary significantly for a given mean wind speed because of the impact of other atmospheric conditions, such as stability. By comparing AEP and ARP gains computed using wind speed-dependent turbulence and a constant turbulence intensity of 9% in FLORIS, we found that the relationship between AEP gains and ARP gains is highly sensitive

to different turbulence intensity modeling approaches. Therefore, future research should seek to include more realistic models of time-varying turbulence. On the other hand, the price time series were derived from recorded hourly prices at the specific locations of the wind plants. So, while there is uncertainty as to the exact level of energy gain from wake steering control that could be produced for a particular wind speed range or hour of the day, the conclusion that higher prices, together with larger relative energy gains from wake steering, during low to medium wind speeds (see Fig. 7) drive higher revenue gain from wake steering relative to energy gain is likely to be robust. Another limitation is that the analysis was performed at a small subset of wind plants, and we saw substantial variation across the plants analyzed. Despite the variation across plants, the general conclusions that revenue gains are larger than energy gains in regions with high wind penetration were robust, and the broad differences in pricing patterns between regions are also likely to be robust across much of each region.

Looking forward, the ability of wake steering control to provide *some* energy gains during low to medium wind speeds suggests the control technology may provide improved value as wind penetration increases and higher prices shift toward lower wind speed hours. Of course, one caveat is that increased energy storage and/or interregional transmission may smooth the price impacts of increased wind penetration. Further study of the interactions between wind farm flow control and market value are of interest, especially regarding control for offshore wind plants. Offshore plants face strong trade-offs when it comes to turbine spacing, in particular due to the expense of leasing offshore development areas and to the expense of connecting distantly spaced turbines; these expenses provide incentive to space offshore turbines relatively close to each other. Together with lower turbulence offshore (Bodini et al., 2019) as well as the trend toward larger wind turbines with higher rated power, these factors increase the likelihood of greater wake losses for offshore wind plants (Pryor et al., 2021), thus enhancing the importance of wind farm flow control strategies such as wake steering.

## Appendix A: Energy and revenue gains with constant turbulence intensity

**Table A1:** Regional averages from 2018 to 2020 of AEP (energy) gain and ARP (revenue) gain, based on FLORIS predictions with a constant turbulence intensity of 9% for all wind speeds, and wind penetration in 2020.

| Electricity Market Region | Energy (AEP) Gain | Revenue (ARP) Gain | ARP Gain / AEP Gain | 2020 Annual Wind Penetration |
|---|---|---|---|---|
| SPP | 0.65% | 0.79% | 1.21 | 31% |
| ERCOT | 0.50% | 0.57% | 1.15 | 23% |
| MISO | 1.22% | 1.31% | 1.07 | 11% |
| PJM | 0.87% | 0.90% | 1.03 | 3% |

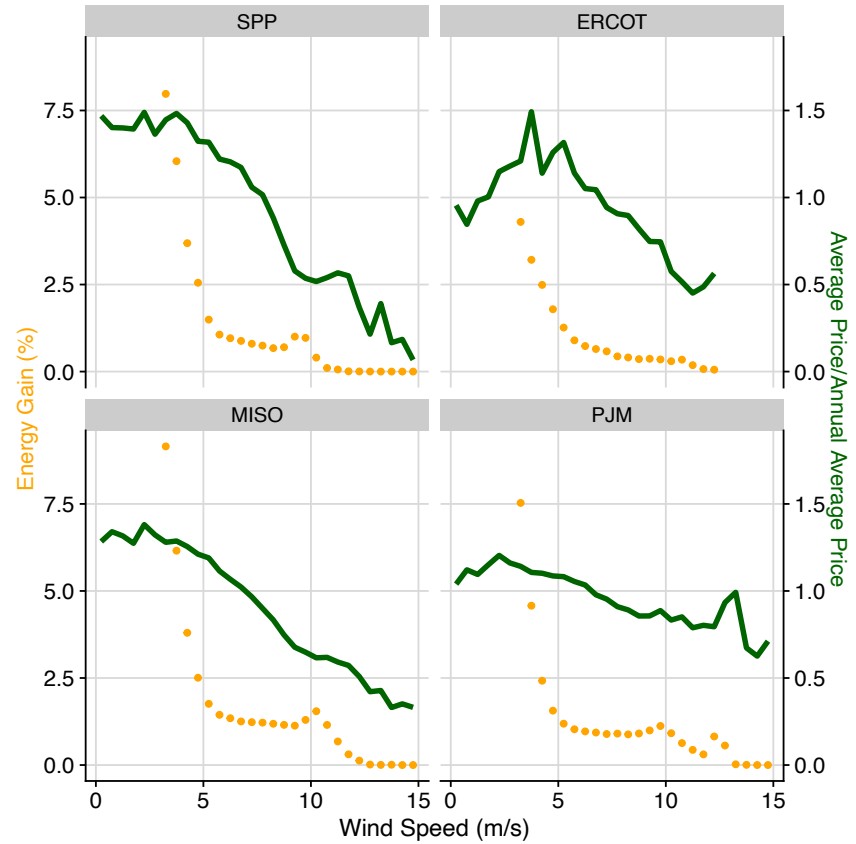

**Figure A1:** Relative energy gains and normalized electricity prices as a function of wind speed for each region, based on FLORIS predictions with a constant turbulence intensity of 9% for all wind speeds. Relative energy gains are calculated independently for each 0.5 m/s wind speed bin. Average prices are also calculated independently for each wind speed bin, but then normalized by the respective annual average price across all wind speeds. Values are averaged across all wind plants in each region and the years 2018–2020.

*Code availability.* The FLORIS code used to model wind plant energy production and optimize wake steering in this paper is available at https://github.com/NREL/floris (NREL, 2022).

*Author contributions.* DM, PF, and ES envisioned the investigation of wake steering value. DM prepared the wind resource
and electricity price data. ES performed the FLORIS modeling and optimization steps. DM and SJ led the analysis of the results, with significant contributions from ES and PF. ES and DM prepared the manuscript with significant contributions from SG and PF.

*Competing interests.* Author Paul Fleming is a member of the editorial board of Wind Energy Science.

*Disclaimer.* The views expressed in the article do not necessarily represent the views of the DOE or the U.S. Government. The
455 U.S. Government retains and the publisher, by accepting the article for publication, acknowledges that the U.S. Government retains a nonexclusive, paid-up, irrevocable, worldwide license to publish or reproduce the published form of this work, or allow others to do so, for U.S. Government purposes.

*Acknowledgements.* The authors thank Patrick Gilman for supporting and engaging with this work, Ryan Wiser for early help in organizing the research team, and Owen Roberts for helpful discussions. A portion of the research was performed using
computational resources sponsored by the Department of Energy's Office of Energy Efficiency and Renewable Energy and located at the National Renewable Energy Laboratory.

*Financial support.* This work was authored in part by the National Renewable Energy Laboratory, operated by Alliance for Sustainable Energy, LLC, for the U.S. Department of Energy (DOE) under contract no. DE-AC36-08GO28308. Funding was provided by the U.S. Department of Energy Office of Energy Efficiency and Renewable Energy Wind Energy Technologies
Office. Support for Lawrence Berkeley National Laboratory was provided by the U.S. Department of Energy's Office of Energy Efficiency and Renewable Energy under Lawrence Berkeley National Laboratory contract no. DE-AC02-05CH11231.

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
