# Peer review of "The value of wake steering wind farm flow control in U.S. energy markets"

_Wind Energy Science, 2023_

## Author Response (AR1)

We thank the two reviewers for their interest in this research and their insightful comments. In particular, we value the reviewers' comments regarding the need to include variable turbulence intensities in the wake steering simulations to better capture the impact of realistic turbulence variations on the predicted energy and revenue gains from wake steering. We believe the suggestions have significantly improved the article. We have provided responses to the specific comments below, and we point out where the article has been modified to address the comments using blue text.

**Reviewer 1 Comments**

The article is scientifically interesting, novel, and of high quality in general, contributing to the interesting discussions on 'beyond power maximization' via flow control. It showcases the potential of wake steering in terms of generating additional income with fluctuating market prices across several sites and market regions in the US. There are a few points the article should be updated on:

1. Reviewer Comment: For the title: The authors are suggested to keep the same taxonomy throughout the article and rephrase wind farm control to wind farm flow control.

   Author response: Thank you for the suggestion. We have updated "wind farm control" to "wind farm flow control" in the title and throughout the article.

2. Reviewer Comment: Entire manuscript: The added value (or even revenue) via wake steering (or WFFC in general) is much more complex than potential additional income. For example, structural loading of (some of) the turbine(s) can increase during wake steering control which might have implications for O&M costs, hence the revenue over asset lifetime. This is briefly discussed in the Introduction section as well.

   Therefore, the authors are encouraged to refrain from over-generalizing potentially and comply with the taxonomy introduced previously by Kölle et al. (2022) https://doi.org/10.5194/wes-7-2181-2022 (e.g., income instead of revenue or value, annual income gain instead of AVP).

   Author response: Thank you for the suggestion. Based on this comment, we have removed the term 'value', since 'value' can sometimes be interpreted as a 'net of cost' gain, whereas we are solely interested in *gross* change to revenue. We now use the terminology "annual revenue of production (ARP)," throughout the paper, which is clearly defined in the text as calculated by summing an annual time series of hourly energy production multiplied by the hourly electricity prices at electricity nodes near each wind plant. There are also several places throughout the paper where we have replaced "value" with phrases that include "revenue" instead. We use "revenue" instead of "income" because income is typically defined as the net of cost gain (i.e., with O&M costs subtracted), whereas we only consider the gross revenue in this study.

   The emphasis on revenue is initially highlighted on line 42 (line 51 in the file with tracked changes): "*value* of the additional energy to the grid at the location and time it is added, quantified as the additional revenue generated."

   The definition of ARP is clarified on line 253 (284 in the file with tracked changes): "A parallel process is followed to determine the gain in annual revenue of production (ARP), which is calculated

by summing annual time series of hourly energy production multiplied by the hourly electricity prices"

and its relationship to the term "income gain" is explained on line 260 (291 in the file with tracked changes): "Note that ARP is similar to the "income gain" reported by Kölle et al. (2022) and the "energy value" (EV) metric presented by Millstein et al. (2021), except ARP is expressed in units of \$/year, whereas EV represents the revenue per unit of energy produced (\$/MWh)."

3.  Reviewer Comment: Abstract: Line 6 - The very first line is slightly misleading as WFFC can have several other objectives than the two listed (e.g., the focus of this article) - consider rephrasing. The first sentence in the Introduction section seems more representative of the technology indeed.

    Author response: This is a good point. There could be other WFFC objectives, such as tracking a reference power signal as opposed to simply maximizing power. To better reflect that power maximization and load reduction are only two examples of WFFC objectives, we have rephrased the sentence to make it more like the first sentence of the introduction:

    Line 7: "Wind farm flow control represents a category of control strategies for achieving wind plant-level objectives, such as increasing wind plant power production and/or reducing structural loads, by mitigating the impact of wake interactions between wind turbines."

4.  Reviewer Comment: Section 3.1: Line 119 - This is not the main point of the article but can be noted here regardless:

    The 'secondary steering' might as well occur due to non-uniform inflow at the downstream turbine, facing a steered wake - especially if/when the implemented FLORIS assumes 'zero-yaw' being the freestream wind direction. A similar effect to secondary steering is in fact observed for the downstream turbines under partial wakes by e.g., Schepers et al. (2012) https://doi.org/10.1002/we.488. There, they report steering of the second wake although all the turbines are under 'normal operation', where the partial wake causes the downstream turbine to effectively misalign.

    So perhaps the 'secondary steering' can (also) be about that effective misalignment further down the row, rather than the upstream steering continuing to deflect the wake(s) downstream?

    Author response: This is an interesting point, and we appreciate the link to the insightful paper by Schepers et al. In previous work using large eddy simulation, we observed that large-scale trailing vortices caused by yaw misalignment contribute to "secondary steering" of wakes at downstream turbines. For example, this is discussed in Fleming et al., "A simulation study demonstrating the importance of large-scale trailing vortices in wake steering," *Wind Energy Science*, 2018, https://doi.org/10.5194/wes-3-243-2018. But as you suggest and as is shown in the Schepers et al. paper (and as is briefly discussed in Section 8 of Fleming et al., "Continued results from a field campaign of wake steering applied at a commercial wind farm – Part 2," *Wind Energy Science*, 2020, https://doi.org/10.5194/wes-5-945-2020), the secondary steering phenomenon may be caused in part by non-uniform inflow or changes in wind direction at the downstream turbine locations.

We have pointed this out on line 121 (139 in the file with tracked changes): "Note that the secondary steering effect could also be partially explained by changes in the effective wind direction at waked wind turbines because of non-uniform inflow (Schepers et al., 2012)."

5.  Reviewer Comment: Section 3.1: Line 129 - TI = 8% for the entire analysis is too simplistic of an assumption and an important deficiency of the study as it might have a significant impact on the reported gains and conclusions.

    Clearly, the FLORIS validation studies mentioned here involve the calibration of the parameters, not the atmospheric variables. Surely there are several reasons why the model predictions are more accurate for that TI range at that site, including the physical limitations of the deficit model implemented. However, TI would indeed change over the investigated sites and the 3-year period, similar to the other atmospheric and economic variables. The temporal and spatial variability of TI should therefore be considered similar to wind direction and speed - ideally also from the same ERA5 simulations (for an example of conversion TKE of WRF runs to TI see https://orbit.dtu.dk/files/274327114/TKE2TI.pdf)

    Author response: Thanks for highlighting this important limitation of in our analysis. And thanks for bringing the report on the conversion of TKE to TI to our attention. This will be useful for our future work. We agree that using a constant TI of 8% may be too simplistic because of potential correlations between turbulence levels (and therefore the effectiveness of wind farm flow control) and electricity prices. We found that electricity prices tend to be higher at low wind speeds, where power gains from wake steering are relatively high when using a fixed TI value, which contributes significantly to the larger value gains compared to AEP gains that we found. However, turbulence is generally higher at lower wind speeds as well, which reduces the power gains from wake steering during high-price low-wind speed periods. This trend has the potential to change the main conclusion of the article if it results in the relative value gains being lower than the relative AEP gains. Because we don't have access to TI or TKE in the ERA5 data we are using, we are now including TI variability in the FLORIS simulations by assigning TI as a function of wind speed, following the Normal Turbulence Model definition from the IEC 61400-1 standard, in which TI = I_ref*(0.75*U + 5.6)/U, where U is the mean wind speed. Although in reality there can be a wide range of TI values for a given wind speed, we feel that this model greatly improves the analysis by capturing the most important TI trend for this study.

    Note that the TI value of 8% we used does not necessarily represent a physical TI of 8%. Instead, we used TI as a tuning variable to match wake losses measured from field data. However, as the other reviewer suggested, 8% may be too low of a value to represent land-based wind plants. After reviewing some of our previous work comparing FLORIS to field measurements, we believe that a constant TI value of 9% represents the average wake losses at land-based wind plants more accurately. Therefore we are tuning "I_ref" in the Normal Turbulence Model formula so the average baseline wake losses for each wind plant match the average wake losses corresponding to a constant TI of 9%.

    As expected, including variable TI in the analysis has changed the results significantly throughout the article. To summarize, the AEP gains are now lower, as expected, because of the decision to tune the turbulence intensity to match average annual wake losses corresponding to the higher TI value of 9%. The relative value gains from wake steering are still higher than the AEP gains, on average, but the ratio between AVP gain and AEP gain is now lower. Whereas AVP gains exceeded AEP gains by

roughly 10% on average in the original manuscript, they now exceed AEP gains by approximately 4% on average. Further, the ratios between the regional average AVP gains and AEP gains were 1.21, 1.14, 1.07, and 1.02 for SPP, ERCOT, MISO, and PJM in the original draft (in decreasing order of wind penetration levels). With the new turbulence modeling approach, the ratios are now reduced to 1.11, 1.09, 1.01, and 1.00, respectively, for the four regions (see Table 2). Thus, although the results are weakened, the main conclusions of the article have not changed: the expected revenue gain still exceeds the expected AEP gain, and the ratio between revenue gain and AEP gain increases as wind penetration increases.

In addition to the numerical results that have changed throughout the article and updated figures (Figs. 4-9), the main changes to the article are as follows.

The discussion of turbulence intensity modeling in FLORIS in Section 3.1 starting on line 131 (149 in the file with tracked changes) has been changed to:

"Wake losses as well as the energy gain possible with wake steering depend strongly on turbulence intensity (Bensason et al., 2021; Simley et al., 2022); both wake losses and the potential increases in wind plant power production from wake steering are higher when turbulence is low. However, the ERA5 wind resource data set used to determine time series of hourly wind speeds and directions for each wind plant in this study (which will be discussed in Section 3.3) does not contain a measure of turbulence. To address this limitation while still capturing the impact of time-varying turbulence intensities in FLORIS, we model turbulence intensity (TI) as a function of wind speed following the normal turbulence model definition in the IEC 61400-1 wind turbine design standard, in which turbulence decreases as wind speed increases (International Electrotechnical Commission, 2005):

$$\mathrm{TI}(\bar{u}) = I_{\mathrm{Ref}}\left(0.75 + \frac{5.6}{\bar{u}}\right), \qquad\qquad (1)$$

where $\bar{u}$ is the mean wind speed—given by the hourly wind speed from the ERA5 data set in this work—and $I_{\mathrm{Ref}}$ is intended to be the expected value of the turbulence intensity at 15 m/s. But in this study, we treat $I_{\mathrm{Ref}}$ as a tuning parameter to achieve desired average annual wake losses at each wind plant. Specifically, Fleming et al. (2020) and Fleming et al. (2021) found that when using a turbulence intensity of 8%–10%, FLORIS predictions closely matched the average wake losses experienced by a pair of wind turbines at commercial wind plants; therefore, we tune $I_{\mathrm{Ref}}$ for each wind plant so that the average annual wake losses predicted using FLORIS match those based on a contact turbulence intensity of 9%."

To highlight the impact that variable turbulence intensities have on the results, we have added Appendix A, which contains a version of Table 2, showing what the regional average AEP and revenue gains would be if we had instead used a constant TI of 9%, and a version of Fig. 7, showing the relative energy gains from wake steering and the normalized prices as a function of wind speed for the four regions. The appendix and the sensitivity of the results to changes in turbulence intensity in general, are discussed in a new paragraph at the end of Section 5.1 on line 316 (353 in the file with tracked changes):

"To reveal the impact that wind speed-dependent turbulence intensity values in FLORIS have on the estimated increases in revenue from wake steering, regional averages of AEP gain and ARP gain over all years investigated are provided in Table A1 in Appendix A for a constant turbulence intensity value of 9% in FLORIS. The AEP gains from wake steering are similar when using variable or constant turbulence intensities, but the ARP gains are significantly higher when assuming a constant turbulence intensity. For a constant turbulence intensity of 9%, the average ARP gain over all wind

plants and years is 10% greater than the average AEP gain, and the ARP gain is 21% larger than the AEP gain for SPP. As will be explained in Section 5.2, the lower ARP gains that result from more realistic wind speed-dependent turbulence intensities in FLORIS can be explained by a weaker correlation between electricity prices and the relative energy gains from wake steering, compared to the scenario with a constant turbulence intensity."

And also in a new paragraph towards the end of Section 5.2 on line 357 (424 in the file with tracked changes):
"The impact of lower turbulence intensity values as wind speed increases (given by Eq. (1)) on the relative energy gain can be clearly seen in Fig. 7 for MISO and PJM for wind speeds between 5 m/s and 10 m/s. In general, during below-rated wind plant operation, as turbulence intensity decreases, wake losses increase, and wake steering becomes more effective at increasing wind plant power production. On the other hand, as shown in Fig. A1 in Appendix A, when using a constant turbulence intensity value of 9% in FLORIS, the relative energy gains from wake steering exhibit a more consistent reduction as wind speed increases for all regions. Consequently, the correlation between relative energy gains and electricity prices is much stronger, resulting in the larger ARP gains compared to AEP gains listed in Table A1 compared to the results with variable turbulence intensity."

Lastly, we updated the discussion of the limitations of the turbulence intensity model in the Discussion and Conclusions section to address how we are now including the dependence between wind speed and TI on line 420 (541 in the file with tracked changes):
"As discussed in Section 3.1, we attempted to capture much of the impact of time-varying turbulence intensity on AEP and ARP gains by assigning turbulence intensity in FLORIS as a function of wind speed, based on the IEC normal turbulence model definition (in which turbulence decreases as wind speed increases); however, while this models the expected average relationship between wind speed and turbulence intensity, turbulence can vary significantly for a given mean wind speed because of the impact of other atmospheric conditions, such as stability. By comparing AEP and ARP gains computed using wind speed-dependent turbulence and a constant turbulence intensity of 9% in FLORIS, we found that the relationship between AEP gains and ARP gains is highly sensitive to different turbulence intensity modeling approaches. Therefore, future research should seek to include more realistic models of time-varying turbulence."

6. Reviewer Comment: Section 3.3: Line 178 - The listed correlations, are they Pearson coefficients? Also, there might be a typo for the second one (weak correlations <0.72?)

   Author response: Yes, these are Pearson coefficients. Also, >0.72 is not a typo, but perhaps not particularly clear.

   We have updated that sentence on line 191 (220 in the file with tracked changes) to specify:
   "Davidson and Millstein (2022) found similar results in Texas, and also showed that Pearson correlation coefficients between modeled and reported hourly generation were strongest during daytime and during winter months (>0.85) and weakest during the summer nighttime (~0.72)."

7. Reviewer Comment: Section 4.1: It seems that the optimal yaw settings per site/per turbine are defined for maximum power objective only (then converted to income). However, it is discussed extensively in the article (also later on) that maximum power does not necessarily correspond to

maximum income. Wouldn't it be more relevant to perform the optimization with the maximum income objective directly?

Author response: This is an interesting point that is worthy of future research. However, in this study we are not including any operations and maintenance or other costs when computing the gross revenue. Including costs associated with wake steering in addition to the revenue gained would require a significant amount of modeling effort, which is out of the scope of this study. Therefore, optimizing for power maximization is equivalent to optimizing for revenue.

This has been clarified on line 223 (253 in the file with tracked changes): "Note that in this study, the optimal yaw offsets for power maximization and revenue maximization are equivalent because we do not consider the potential impacts of wake steering on expenditures such as operations and maintenance costs during the lifetime of the wind plants."

8.  Reviewer Comment: Section 4.1: Line 226 - "... if an optimal yaw offset of 12.5° is found during the coarse search for a particular turbine, offsets of 6.25°, 9.375°, 12.5°, 15.625°, and 18.75° will be evaluated during the refined search..."

    How realistic is it to yaw a turbine 6.25deg or even 9.375 degrees? Wouldn't it be more feasible (both for potential field implementation and for the grid search) to use round steps for yaw?

    Author response: We believe it is realistic to implement wake steering in the field using precise, non-round yaw offset commands. During our experience with wake steering field campaigns (including one in which the wind turbine OEM is involved), this has not been an issue. As far as the specific non-round offsets selected for the coarse and refined searches, they are simply the result of evaluating offsets at even intervals between the yaw offset bounds (+/-25 degrees in this case). However, as discussed near the end of Section 4.1, we actually use an additional refinement step after the Serial-Refine (SR) process, in which the resulting offsets from SR (e.g., 6.25°, 9.375°, 12.5°, etc.) are used as initial conditions for an optimization using the gradient-based sequential least-squares programming (SLSQP) method, limited to 10 iterations to reduce computation time. Therefore, the final yaw offsets that are found can take any value between +/- 25 degrees, not just the discrete points searched in the SR method.

9.  Reviewer Comment: Section 5.1: There is another important assumption to be discussed here, translating the potential AEP gains into income gains. It is that the analysis assumes the wind power plants are 'price takers', not 'price makers' - meaning that the additional increase in wind power production via wake steering does not affect the electricity price. However, as the authors indicated earlier, the prices are indeed highly correlated with wind production (not the wind speed directly), especially in grids with high share of wind. So if all the WFs in the region apply wake steering simultaneously and increase wind energy production on a large scale, would the price stay the same?

    Author response: This is a subtle but correct point. If all wind plants apply wake control, it might slightly increase fleet output during those hours and thus provide a negative feedback effect to the wind plants. However, even in that situation, wake control would increase energy output from the fleet by a relatively small amount, whereas market price changes are sensitive to large swings in fleet output. Secondly, it is unlikely that all plants in a region would face the same wake control opportunities at the same time, dampening any potential feedback effects on prices.

10. Reviewer Comment: Section 5.2: Line 287 - The true correlation lies with the high prices that are typically observed during low wind production (at least for the investigated period 2018-2020, which might look different in a future market scenario)

   Author response: This is an interesting point, there are multiple trends happening here, prices are much more likely to spike when wind power is limited, as the reviewer pointed out. Separately however, negative or near zero prices are also associated with high output from the wind fleet (see the ReWEP tool for example: https://emp.lbl.gov/renewables-and-wholesale-electricity-prices-rewep). We think there is good reason to believe these price trends, which drive the slightly higher income gains than energy gains will likely strengthen in the future with additional wind deployment. However, there are possible changes to electricity grid which might counteract those trends – mainly increased transmission and increased storage options.

   This last now point is now discussed in the article on line 438 (561 in the file with tracked changes): "Of course, one caveat is that increased energy storage and/or interregional transmission may smooth the price impacts of increased wind penetration."

11. Reviewer Comment: Section 5.2: Line 294 - As the prices are connected to production rather than the wind speeds as discussed earlier, the authors are encouraged to rephrase the opening argument here so that the wake steering contribution during high electricity prices is highlighted (not necessarily during low wind speeds).

   This is clarified nicely later in the paragraph.

   Author response: We have clarified this point in the sentence beginning on line 333 (384 in the file with tracked changes): "This observation matches the hypothesis that controls to increase power production at low wind speeds are more valuable (on a per-megawatt-hour basis) than controls at higher wind speeds (because low wind speeds are correlated with higher energy prices and high wind speeds are correlated with low energy prices)."

12. Reviewer Comment: Section 5.3: Figure 8 - The 'Ratio' presented here needs a bit more elaboration. For the y-axis, it might be more helpful to list the ratio in percentages as well as to label it with the equation already in the plot (i.e. AEP/AEPmax?) - especially since it seems to be different than the Price Ratio in Figure 7.

   Author response: This is a good point, we have changed the y-axis label "Ratio" in Figure 8 to "(AEP or ARP gain)/(total AEP or ARP gain)" to clarify the values plotted.

13. Reviewer Comment: Section 5.4:  Lines 330-335 -  For the concentration analysis and discussion, perhaps a histogram or CDF plot would be clearer? Log-scale does not seem to indicate significant differences between energy and income gain trends, at least not at first look.

   Author response: Yes, we agree that one does need to look very closely at the original Figure 9 to see the difference and that the figure was hard to interpret. We plotted it as a histogram, as suggested, but it continued to be hard to interpret, so we removed the figure and instead rely on the original Fig. 10 (now Fig. 9) to support the discussion about the concentration of revenue gain in

a small number of hours. Section 5.4 has been rewritten to remove text that referred to the original Fig. 9 (no major additions to the text were made).

14. Reviewer Comment: Figures 5 to 9: It would be helpful to the reader with a grid to follow the values the curves (or data points) correspond to on each axes.

    Author response: Thank you for the suggestion. We have added grids to Figures 5 through 9.

Although it might take some time, I believe the analysis and conclusions presented in the article would be much stronger with better TI estimation per site, as well as price-driven optimization (with more feasible controller settings) for the income gain analysis.

Author response: Thank you for the careful review of the article. We believe that the improved TI modeling has made the article significantly stronger.

Thanks and greetings,

Tuhfe Göcmen

**Reviewer 2 Comments**

General comments

The paper addresses the impact of wake steering wind farm flow control on potential revenue gains. As this aspect of wind farm flow control has not been studied extensively so far, this paper is a very relevant contribution to the existing literature. The results presented in the paper indicate an increase in annual revenue value compared to increased energy production with wake steering. This conclusion could provide additional incentive for operators to implement wind farm flow control on their wind farms.

The paper is well written and has a clear structure. The approach taken by the authors is straightforward and sensible. A few comments, which I believe could improve the quality of the paper, are provided below:

Author response: Thank you for your interest in this work and for your time reviewing the article. We're glad that you believe the article is a very relevant contribution to the existing literature. We believe we have addressed your insightful comments as explained in the following responses.

Specific comments
1. Reviewer Comment: Section 3.1: Can the authors explain their choice for the Gauss-curl hybrid wake model over the cumulative-curl wake model (Bay et al., 2023)? The latter was found to be more accurate in modelling wake losses for large wind farms and is included in FLORIS. Do you expect large differences in the results when switching to a different model?

Author response: We are using the Gauss-curl hybrid (GCH) model in this study because we have performed several wake steering experiments at commercial wind plants in which the predicted power gains from wake steering were found to match the measured power gains reasonably well (albeit for only one or two controlled turbines):

Fleming et al., "Continued results from a field campaign of wake steering applied at a commercial wind farm – Part 2," *Wind Energy Science*, 2020, https://doi.org/10.5194/wes-5-945-2020.

Simley et al., "Results from a wake-steering experiment at a commercial wind plant: investigating the wind speed dependence of wake-steering performance," *Wind Energy Science*, 2021, https://doi.org/10.5194/wes-6-1427-2021.

Fleming et al., "Experimental results of wake steering using fixed angles," *Wind Energy Science*, 2021, https://doi.org/10.5194/wes-6-1521-2021.

Therefore, we have relatively high confidence in the GCH model for wake steering. But as you point out, the cumulative-curl (CC) model improves the estimation of wake losses for large offshore arrays compared to the GCH model. It may better represent wake losses at large land-based wind plants as well. However, we have not validated the wake steering model in the CC model using field data, so we don't have confidence in its ability to predict power gains from wake steering yet. Therefore, we currently prefer to use GCH to model wake steering. Note that in the future, we expect to use the new Empirical Gaussian model in FLORIS (https://nrel.github.io/floris/empirical_gauss_model.html#empirical-gauss-model) to model wake steering. This model, which is based on the Gaussian wake model, is designed to be easy to tune to field data and captures some of the important wake phenomena for large wind plants, such as the reduced wake recovery for wake interactions over large distances. More validation studies are required before we will be ready to use this model for wake steering modeling, however.

As far as whether we expect large differences in wake steering results when switching to a new model such as the CC model, some initial research comparing the GCH and CC models shows very large discrepancies in the predicted AEP gains. For example, in the report "Wind Farm Control and Layout Optimization for U.S. Offshore Wind Farms" (https://nationaloffshorewind.org/projects/wind-farm-control-and-layout-optimization-for-u-s-offshore-wind-farms/), the AEP gains from wake steering for the CC model were found to be significantly lower than the gains predicted with the GCH model for offshore arrays (see Fig. 9). Consequently, we feel the need to validate the CC model for wake steering using field data before relying on its predictions of AEP gains.

2. Reviewer Comment: Section 3.1: While turbulence levels of 8% might occur at each of the selected wind plants, it seems to be a relatively low level to consider as the average turbulence intensity for onshore purposes. Higher turbulence levels might result in more accurate estimations of the AEP and AVP benefits, which will probably decrease due to the improved wake recovery from higher turbulence. Furthermore, as turbulence intensity is related to wind speed, this should also be considered in the FLORIS simulations.

Author response: This is a great point, and we agree that an average TI of 8% is probably too low for most land-based wind plants. However, instead of representing a physical TI of 8%, the constant TI value of 8% in FLORIS was meant to match the average wake losses and power gains from wake

steering that have been observed using field data from wake steering experiments at commercial wind plants. In other words, we treat TI as a tuning variable that can be tuned so that FLORIS results match observed wake losses or power gains from field data. But your comment made us revisit the choice of 8% as the best TI value for land-based wind plants. After reviewing our previous work comparing FLORIS to field measurements, we believe that a slightly higher constant TI value of 9% represents the average wake losses at land-based wind plants more accurately when using the GCH model. Specifically, the following papers support this choice of constant TI to match the observed wake losses and power gains from wake steering from field experiments at commercial wind plants:

Fleming et al., "Continued results from a field campaign of wake steering applied at a commercial wind farm – Part 2," *Wind Energy Science*, 2020, https://doi.org/10.5194/wes-5-945-2020.

Fleming et al., "Experimental results of wake steering using fixed angle," *Wind Energy Science*, 2021, https://doi.org/10.5194/wes-6-1521-2021.

We also agree that using a constant TI of 8% was an important limitation in our analysis because of potential correlations between turbulence levels (and therefore the effectiveness of wind farm flow control) and electricity prices. We found that electricity prices tend to be higher at low wind speeds, which contributes significantly to the larger value gains compared to AEP gains because wake steering was also found to increasing power more at low wind speeds when using a fixed TI of 8%. However, turbulence is generally higher at lower wind speeds as well, which reduces the power gains from wake steering during high-price low-wind speed periods. This trend has the potential to change the main conclusion of the article if it results in the relative value gains being lower than the relative AEP gains. Because we don't have access to TI (or even TKE) in the ERA5 data we are using, we are now including TI as a function of wind speed in the FLORIS simulations, as you suggest, using the Normal Turbulence Model definition from the IEC 61400-1 standard, in which TI = I_ref*(0.75*U + 5.6)/U, where U is the mean wind speed. Although in reality there can be a wide range of TI values for a given wind speed, we feel that this model greatly improves the analysis by capturing the most important TI trend for this study.

Because our previous research suggests that a TI of 9% in FLORIS matches the average wake losses and power gains from wake steering at land-based wind plants reasonably well, we are tuning "I_ref" in the Normal Turbulence Model formula so the average baseline wake losses for each wind plant match the average wake losses corresponding to a constant TI of 9%.

As expected, including variable TI in the analysis has changed the results significantly throughout the article. To summarize, the AEP gains are now lower, as expected, because of the decision to tune the turbulence intensity to match average annual wake losses corresponding to the higher TI value of 9%. The relative value gains from wake steering are still higher than the AEP gains, on average, but the ratio between AVP gain and AEP gain is now lower. Whereas AVP gains exceeded AEP gains by roughly 10% on average in the original manuscript, they now exceed AEP gains by approximately 4% on average. Further, the ratios between the regional average AVP gains and AEP gains were 1.21, 1.14, 1.07, and 1.02 for SPP, ERCOT, MISO, and PJM in the original draft (in decreasing order of wind penetration levels). With the new turbulence modeling approach, the ratios are now reduced to 1.11, 1.09, 1.01, and 1.00, respectively, for the four regions (see Table 2). Thus, although the results are weakened, the main conclusions of the article have not changed: the expected revenue gain still exceeds the expected AEP gain, and the ratio between revenue gain and AEP gain increases as wind penetration increases.

In addition to the numerical results that have changed throughout the article and updated figures (Figs. 4-9), the main changes to the article are as follows.

The discussion of turbulence intensity modeling in FLORIS in Section 3.1 starting on line 131 (149 in the file with tracked changes) has been changed to:

"Wake losses as well as the energy gain possible with wake steering depend strongly on turbulence intensity (Bensason et al., 2021; Simley et al., 2022); both wake losses and the potential increases in wind plant power production from wake steering are higher when turbulence is low. However, the ERA5 wind resource data set used to determine time series of hourly wind speeds and directions for each wind plant in this study (which will be discussed in Section 3.3) does not contain a measure of turbulence. To address this limitation while still capturing the impact of time-varying turbulence intensities in FLORIS, we model turbulence intensity (TI) as a function of wind speed following the normal turbulence model definition in the IEC 61400-1 wind turbine design standard, in which turbulence decreases as wind speed increases (International Electrotechnical Commission, 2005):

$$\text{TI}(\bar{u}) = I_{\text{Ref}}\left(0.75 + \frac{5.6}{\bar{u}}\right), \tag{1}$$

where $\bar{u}$ is the mean wind speed—given by the hourly wind speed from the ERA5 data set in this work—and $I_{\text{Ref}}$ is intended to be the expected value of the turbulence intensity at 15 m/s. But in this study, we treat $I_{\text{Ref}}$ as a tuning parameter to achieve desired average annual wake losses at each wind plant. Specifically, Fleming et al. (2020) and Fleming et al. (2021) found that when using a turbulence intensity of 8%–10%, FLORIS predictions closely matched the average wake losses experienced by a pair of wind turbines at commercial wind plants; therefore, we tune $I_{\text{Ref}}$ for each wind plant so that the average annual wake losses predicted using FLORIS match those based on a contact turbulence intensity of 9%."

To highlight the impact that variable turbulence intensities have on the results, we have added Appendix A, which contains a version of Table 2, showing what the regional average AEP and revenue gains would be if we had instead used a constant TI of 9%, and a version of Fig. 7, showing the relative energy gains from wake steering and the normalized prices as a function of wind speed for the four regions. The appendix and the sensitivity of the results to changes in turbulence intensity in general, are discussed in a new paragraph at the end of Section 5.1 on line 316 (355 in the file with tracked changes):

"To reveal the impact that wind speed-dependent turbulence intensity values in FLORIS have on the estimated increases in revenue from wake steering, regional averages of AEP gain and ARP gain over all years investigated are provided in Table A1 in Appendix A for a constant turbulence intensity value of 9% in FLORIS. The AEP gains from wake steering are similar when using variable or constant turbulence intensities, but the ARP gains are significantly higher when assuming a constant turbulence intensity. For a constant turbulence intensity of 9%, the average ARP gain over all wind plants and years is 10% greater than the average AEP gain, and the ARP gain is 21% larger than the AEP gain for SPP. As will be explained in Section 5.2, the lower ARP gains that result from more realistic wind speed-dependent turbulence intensities in FLORIS can be explained by a weaker correlation between electricity prices and the relative energy gains from wake steering, compared to the scenario with a constant turbulence intensity."

And also in a new paragraph towards the end of Section 5.2 on line 357 (424 in the file with tracked changes):

"The impact of lower turbulence intensity values as wind speed increases (given by Eq. (1)) on the relative energy gain can be clearly seen in Fig. 7 for MISO and PJM for wind speeds between 5 m/s and 10 m/s. In general, during below-rated wind plant operation, as turbulence intensity decreases, wake losses increase, and wake steering becomes more effective at increasing wind plant power production. On the other hand, as shown in Fig. A1 in Appendix A, when using a constant turbulence intensity value of 9% in FLORIS, the relative energy gains from wake steering exhibit a more consistent reduction as wind speed increases for all regions. Consequently, the correlation between relative energy gains and electricity prices is much stronger, resulting in the larger ARP gains compared to AEP gains listed in Table A1 compared to the results with variable turbulence intensity."

Lastly, we updated the discussion of the limitations of the turbulence intensity model in the Discussion and Conclusions section to address how we are now including the dependence between wind speed and TI on line 420 (541 in the file with tracked changes):
"As discussed in Section 3.1, we attempted to capture much of the impact of time-varying turbulence intensity on AEP and ARP gains by assigning turbulence intensity in FLORIS as a function of wind speed, based on the IEC normal turbulence model definition (in which turbulence decreases as wind speed increases); however, while this models the expected average relationship between wind speed and turbulence intensity, turbulence can vary significantly for a given mean wind speed because of the impact of other atmospheric conditions, such as stability. By comparing AEP and ARP gains computed using wind speed-dependent turbulence and a constant turbulence intensity of 9% in FLORIS, we found that the relationship between AEP gains and ARP gains is highly sensitive to different turbulence intensity modeling approaches. Therefore, future research should seek to include more realistic models of time-varying turbulence."

3. Reviewer Comment: Section 3.1: The turbines are modelled after the IEA 3.4MW RWT, resulting in a relatively low rated wind speed compared to real-life wind turbines of similar size. As I understand, these usually have a rated wind speed between 11-14 m/s. How will this low rated wind speed have affected the AEP calculations, as you are essentially cutting the effective wind farm flow control range?

Author response: Although the turbines in the study are modeled after the IEA 3.4 MW turbine, the power and thrust curves are adjusted to match the estimated rated wind speed for each turbine, as described in Section 3.2. This process assumes that the rated wind speed of the turbines is equal to the rated wind speed of the IEA 3.4 MW reference turbine (9.8 m/s) multiplied by the cube root of the ratio of the specific power (W/m^2) of the turbine of interest to the specific power of the IEA 3.4 MW turbine (this relationship is revealed by simplifying the ratio of the estimated rated wind speed in Equation 2 for the two turbines). Therefore, turbines with higher specific powers will be modeled as having higher rated wind speeds. For the turbines in the study, the estimated rated wind speeds range from 9 m/s to 10.5 m/s, with a mean value of 9.6 m/s.

We now mention these values on line 180 (208 in the file with tracked changes) to highlight the different rated wind speeds: "Across all 15 wind plants, the estimated mean wind speeds of the FLORIS wind turbine models using Eq. (3) range from 9 m/s to 10.5 m/s, with a mean value 9.6 m/s."

However, as you point out, the published rated wind speeds for commercial wind turbines like the ones modeled in this study are typically higher than these estimated values. This may in part be due to the manufacturer-provided power curves including the effects of turbulence during the averaging

periods used to create the power curves, which would smooth out the knee of the power curve and delay the wind speed at which the turbine reaches rated power to a higher wind speed. Since we do not have access to the power curves or thrust curves for the wind turbines at the wind plants analyzed in this study, we instead approximate the power and thrust curves by adjusting the curves for the IEA 3.4 MW reference turbine. We acknowledge that this is a simplification of the study, and the results could change if we used the true power and thrust curves for the turbines. We expect that if we used power and thrust curves with higher rated wind speeds, the AEP gains would be higher since wake losses and thus wake steering gains would occur over a wider range of wind speeds.

We now comment on the uncertainty in AEP gain resulting from uncertainty in the coefficient of power and thrust curves on line 407 (528 in the file with tracked changes) in the discussion and conclusions section: "Imperfections in the FLORIS wake models as well as uncertainty in the estimated wind turbine coefficients of power and thrust as a function of wind speed (see Section 3.2) create uncertainty in the predicted AEP and ARP gains."

4.  Reviewer Comment: Section 3.3/4.2: To my understanding, in the presented framework FLORIS only predicts the hourly power generation for a single wind direction. A more accurate assessment of the hourly production can be obtained by evaluating FLORIS for multiple wind directions in combination with a Gaussian, or similar, distribution. This could have a negative impact on the estimated AEP and AVP. Can the authors comment on this, do you expect large differences if such an approach was considered?

    Author response: This is a very interesting point. You are correct that for each hourly timestep the baseline and optimized power in FLORIS are computed for the single hourly wind speed and direction from the ERA5 time series. In reality, the wind direction would vary within the hourly period. In general, we don't believe that this would have a negative impact on the estimated AEP and AVP, however, because the turbines would also change their yaw orientations within an hourly period to adapt to the new wind directions, thus implementing the optimal yaw offsets for the range of wind directions encountered. (Note that this is assuming the turbines' yaw positions can perfectly track the varying wind directions – see the discussion in the next paragraph.) There would likely be some cases when the increase in energy or value for a particular timestep would be reduced when a distribution of wind directions is included, for example, if the mean wind direction from the ERA5 timeseries yields the greatest energy gain (e.g., because rows of turbines are roughly aligned with the wind direction) and any deviations from that wind direction produce lower energy gains. But there would also likely be cases when the energy and value gains would be higher when considering a distribution of wind directions, for example if wake steering is not particularly beneficial at the mean wind direction for the timestep, but small deviations from that wind direction produce larger energy gains. We believe these effects would tend to cancel out over one year, yielding AEP and AVP gains close to the values predicted assuming a single wind direction each timestep.

    On the other hand, the variability of the wind direction can have a negative impact on AEP and AVP gain because, in practice, the turbines aren't able to adjust their yaw position fast enough to keep up with the changing wind directions perfectly. Because of the relatively slow yaw controller dynamics, the turbines' yaw orientations will tend to lag the optimal yaw positions. This phenomenon has been evaluated previously by several different authors (see Simley et al., "Design and analysis of a wake steering controller with wind direction variability," *Wind Energy Science*, 2020, https://doi.org/10.5194/wes-5-451-2020 and references therein for details). However, in the

present study, our goal is to assess the *potential* impact of wake steering on AEP and value gain, rather than model turbine-specific controller limitations, which may change as wake steering matures. Therefore, we have not included wind direction variability in the FLORIS predictions.

We now discuss these points at the end of Section 4.2, on line 263 (294 in the file with tracked changes):

"By using hourly wind data from ERA5 to assess wake steering in this study, we are missing information about higher frequency variations in wind speed and direction, which could impact the potential increase in energy from wake steering for any particular hour. For example, if the distribution of wind directions within a specific hour leads to fewer wake interactions than the single hourly wind direction, the energy gain could be lower for that hour. But if the actual distribution of wind directions causes greater wake losses than the single hourly wind direction, the energy gain from wake steering could be enhanced. Therefore, we expect that the impacts of sub-hourly variations in wind conditions will tend to cancel out over the annual periods investigated without significantly affecting the results.
    Note that, in practice, the energy and revenue gains from wake steering are expected to be lower than those predicted by FLORIS because of the inability of wind turbines to perfectly adapt their yaw offsets to the optimal values in realistic time-varying wind conditions (see Simley et al. (2020) and Fleming et al. (2020) for a deeper discussion of this limitation). However, in the present study, our goal is to assess the potential impact of wake steering on AEP and ARP gain, rather than model specific controller limitations, which we expect to be turbine-specific and likely to improve as wake steering technology matures."

5. Reviewer Comment: Section 5.3: "… wind generation tends to be more concentrated in nighttime than daytime." Since atmospheric stability is not taken into account, how is this effect modelled in the current study? Is it only due to less variation in the hourly wind directions?

Author response: This statement was meant to reflect how wind speeds, and thus wind energy generation, are typically higher at night. To clarify this, we added to the sentence on line 372 (439 in the file with tracked changes): "Absolute energy gain peaks at night because in most locations wind generation tends to be more concentrated in nighttime, when wind speeds are typically higher, than daytime."

However, now that we are including TI as a function of wind speed, TI tends to be lower at night as well because wind speeds are higher. Because wake steering is more effective at increasing power production when turbulence is low (all else being equal), we have changed the sentence on line 374 (441 in the file with tracked changes) to: "Additionally, because wind speeds are generally higher at night, the modeled turbulence intensities using Eq. (1) tend to be lower, which yields larger increases in power production from wake steering."

Technical corrections

6. Reviewer Comment: Line 106: "using a simple wind turbine model consisting of the turbines' power and coefficient of thrust curves." Should this be "using a simple wind turbine model consisting of the turbines' power and thrust curves"?

Author response: FLORIS explicitly requires the coefficients of power and thrust as a function of wind speed as part of the wind turbine model. In practice, given the rotor diameter, the coefficients could be determined from the actual power and thrust curves. So the required inputs for FLORIS could be calculated using either the coefficients of power and thrust or the absolute power and thrust curves.

However, since mixing absolute power and the coefficient of thrust in this sentence is confusing, we have changed the sentence beginning on line 106 (124 in the file with tracked changes) to "FLORIS models wind plant power production for different inflow conditions (i.e., wind speed, wind direction, turbulence intensity, air density, wind shear, and wind veer) using a simple wind turbine model consisting of the turbines' coefficients of power and thrust as a function of wind speed, hub heights (the height of the top of the tower supporting the wind turbine), and rotor diameters."

7. Reviewer Comment: Figure 9: Increasing the axis font size will improve readability.

   Author response: The original Figure 9 has been removed from the paper because it is difficult to interpret, as pointed out by Reviewer 1's comment 13. We also felt that the points we wanted to make could be made using the original Figure 10 (now Figure 9).

8. Reviewer Comment: Figure 10: The resolution of this figure is quite low.

   Author response: The resolution of this figure (now Figure 9) has been improved.